# Explore Aggressively, Update Conservatively: Stochastic Extragradient Methods with Variable Stepsize Scaling

**Yu-Guan Hsieh**
Univ. Grenoble Alpes, LJK
38000 Grenoble, France
yu-guan.hsieh@univ-grenoble-alpes.fr

**Franck Iutzeler**
Univ. Grenoble Alpes, LJK
38000 Grenoble, France
franck.iutzeler@univ-grenoble-alpes.fr

**Jérôme Malick**
CNRS, LJK
38000 Grenoble, France
jerome.malick@univ-grenoble-alpes.fr

**Panayotis Mertikopoulos**
Univ. Grenoble Alpes, CNRS, Inria, Grenoble INP, LIG
38000 Grenoble, France
Criteo AI Lab, France
panayotis.mertikopoulos@imag.fr

## Abstract

Owing to their stability and convergence speed, extragradient methods have become a staple for solving large-scale saddle-point problems in machine learning. The basic premise of these algorithms is the use of an extrapolation step before performing an update; thanks to this exploration step, extragradient methods overcome many of the non-convergence issues that plague gradient descent/ascent schemes. On the other hand, as we show in this paper, running vanilla extragradient with stochastic gradients may jeopardize its convergence, even in simple bilinear models. To overcome this failure, we investigate a double stepsize extragradient algorithm where the exploration step evolves at a more aggressive time-scale compared to the update step. We show that this modification allows the method to converge even with stochastic gradients, and we derive sharp convergence rates under an error bound condition.

## 1   Introduction

A major obstacle in the training of generative adversarial networks (GANs) is the lack of an implementable, strongly convergent method based on stochastic gradients. The reason for this is that the coupling of two (or more) neural networks gives rise to behaviors and phenomena that do not occur when minimizing an *individual* loss function, irrespective of the complexity of its landscape. As a result, there has been significant interest in the literature to codify the failures of GAN training, and to propose methods that could potentially overcome them.

Perhaps the most prominent of these failures is the appearance of cycles [4, 6, 7, 20, 21] and, potentially, the transition to aperiodic orbits and chaos [3, 8, 28, 29, 31]. Surprisingly, non-convergent phenomena of this kind are observed even in very simple saddle-point problems such as two-dimensional, unconstrained bilinear games [4, 7, 21]. In view of this, it is quite common to examine the convergence (or non-convergence) of a gradient training scheme in bilinear models before applying it to more complicated, non-convex/non-concave problems.

A key observation here is that the non-convergence of standard gradient descent-ascent methods in bilinear saddle-point problems can be overcome by incorporating a "gradient extrapolation" step before performing an update. The resulting algorithm, due to Korpelevich [13], is known as the *extragradient* (EG) method, and it has a long history in optimization; for an appetizer, see Facchinei & Pang [5], Juditsky et al. [11], Nemirovski [26], Nesterov [27], and references therein. In particular, the extragradient algorithm converges for all pseudomonotone variational inequalities (a large problem class that contains all bilinear games, cf. [13]), and the time-average of the generated iterates achieves an $\mathcal{O}(1/t)$ rate of convergence in monotone problems [26].

The above concerns the application of extragradient methods with perfect, *deterministic* gradients and a non-vanishing stepsize. By contrast, in the type of saddle-point problems that are encountered in machine learning (GANs, robust reinforcement learning, etc.), there are two important points to keep in mind: First, the size of the datasets involved precludes the use of full gradients (for more than a few passes at least), so the method must be run with *stochastic* gradients instead. Second, because the landscapes encountered are not convex-concave, the method's last iterate is typically preferred to its time-average (which offers no tangible benefits when Jensen's inequality no longer applies). We are thus led to the following questions: (*i*) *are the superior last-iterate convergence properties of the EG algorithm retained in the stochastic setting?* And, if not, (*ii*) *is there a principled modification that would restore them?*

**Our contributions.** To motivate our analysis, we first analyse a counterexample to show that the last iterate of stochastic EG fails to converge, even in bilinear min-max problems where deterministic EG methods converge from any initialization. We then consider a class of *double stepsize extragradient* (DSEG) methods with an exploration step evolving more aggressively than the update step and prove it enjoys better convergence guarantees than standard EG in stochastic problems. In more detail:

1. We show that the DSEG algorithm converges with probability 1 in a large class of problems that contains all monotone saddle-point problems.

2. We derive explicit convergence rates for the algorithm's last iterate under an error bound condition. This is the first time that such condition is considered in the analysis of stochastic EG methods, albeit its popularity in the optimization community.

3. For bilinear min-max problems in particular, our analysis establishes that stochastic DSEG methods converge at a $\mathcal{O}(1/t)$ rate. Prior to our work, last-iterate convergence rate for bilinear min-max games had only been studied in the deterministic setting.[1]

4. To account for non-monotone problems, we also provide local versions of these results that hold with (arbitrarily) high probability. Importantly, thanks to the use of a local error bound condition, we can obtain local convergence rates even if the Jacobian at a solution contains purely imaginary eigenvalues.

**Related works.** The approaches that have been explored in the literature to ensure the convergence of stochastic first-order methods, in monotone problems and beyond, include variance reduction with increasing batch size and schemes with vanishing regularization (or "anchoring"). In regard to the former, Iusem et al. [10] showed that using increasing batch size can ensure convergence in pseudomonotone variational inequalities. As for the latter, Koshal et al. [14] and Ryu et al. [30] regularized the problem via the addition of a strongly monotone term with vanishing weight; by properly controlling the weight reduction schedule of this regularization term, it is possible to show the method's convergence in monotone problems.

In contrast to the above, our approach is based on a modification of the choice of the stepsizes, which has only been studied theoretically in the deterministic setting. Zhang & Yu [35] recently examined the convergence of several gradient-based algorithms in unconstrained zero-sum bilinear games with deterministic oracle feedback. Interestingly, they show that the optimal (geometric) rate of convergence in bilinear games is recovered for asymptotically large "exploration" parameters $\gamma \to \infty$ and infinitesimally small "update" parameters $\eta \to 0$. Even though the setting there is quite

| | | Assumption | Guarantee | Rate |
|---|---|---|---|---|
| Extragradient (Mirror-prox) | [11] | monotone | ergodic | $1/\sqrt{t}$ |
| | [12] | strongly monotone | last | $1/t$ |
| | [21] | strictly coherent | last | asymptotic |
| Increasing batch size | [10] | pseudo-monotone | best | $1/\sqrt{t}$ |
| | | | last | asymptotic |
| Repeated sampling | [23] | monotone | ergodic | $1/\sqrt{t}$ |
| SVRE | [2] | strongly monotone + finite sum | last | $e^{-\rho t}$ |
| Double stepsize | Ours | variational stability (VS) | last | asymptotic |
| | | VS + error bound | last | $1/t^{1/3}$ |
| | | monotone + affine | last | $1/t$ |

**Table 1:** Summary of known convergence results of stochastic extragradient methods. For ergodic, last iterate and best iterate guarantees, the convergence metrics are respectively dual gap, squared distance to the solution set and squared residual. Results for single-call [9, 16] and non-extragradient methods [14, 17, 30] are not included.

different from our own, it is interesting to note that the principle of a smaller update stepsize also applies in their case – see also Liang & Stokes [15] and Mishchenko et al. [23] for a concurrent series of results, and Ryu et al. [30] for an empirical investigation into the stochastic setting.

Regarding convergence counterexamples, in a recent paper, Chavdarova et al. [2] showed that if EG is run with a *constant* stepsize and noise with *unbounded* variance, the method's iterates actually diverge at a geometric rate. Motivated by this, they proposed a SVRG-type variance reduced EG method for finite-sum problems and proved a geometric convergence of the algorithm when the involved operator is strongly monotone. Compared to this situation, our counterexample illustrates that the non-convergence persists for *any* error distribution with positive variance (no matter how small) and *any* stepsize sequence (constant, decreasing, or otherwise). In particular, if EG is run with noisy feedback, its trajectories remain non-convergent even if the noise is almost surely bounded and a vanishing stepsize schedule is employed.

Finally, to make our paper's position clear with respect to the large corpus of work on stochastic EG methods, we further provide an overview of the most relevant results in Table 1 and refer the interested reader to the supplement for further discussion.

## 2 Preliminaries

In this section, we briefly review some basics for the class of problems under consideration – namely, saddle-point problems and the associated vector field formulation.

**Saddle-point problems.** The flurry of activity surrounding the training of GANs has sparked renewed interest in saddle-point problems and zero-sum games. To define this class of problems formally, consider a value function $\mathcal{L}\colon \mathbb{R}^{d_1} \times \mathbb{R}^{d_2} \to \mathbb{R}$ which assigns a cost of $\mathcal{L}(\theta, \phi)$ to a player controlling $\theta \in \mathbb{R}^{d_1}$, and a payoff of $\mathcal{L}(\theta, \phi)$ to a player choosing $\phi \in \mathbb{R}^{d_2}$. Then, the *saddle-point problem* associated to a $\mathcal{L}$ consists of finding a profile $(\theta^\star, \phi^\star) \in \mathbb{R}^{d_1} \times \mathbb{R}^{d_2}$ such that, for all $\theta \in \mathbb{R}^{d_1}$, $\phi \in \mathbb{R}^{d_2}$, we have:

$$\mathcal{L}(\theta^\star, \phi) \leq \mathcal{L}(\theta^\star, \phi^\star) \leq \mathcal{L}(\theta, \phi^\star). \tag{SP}$$

In this setting, the pair $(\theta^\star, \phi^\star)$ is called a (global) *saddle point* of $\mathcal{L}$ – or, in game-theoretic terminology, a *Nash equilibrium* (NE). For concision and generality, we will often abstract away from $\theta$ and $\phi$ by setting $x = (\theta, \phi) \in \mathbb{R}^d$ (where, in obvious notation, $d = d_1 + d_2$).

**Vector field formulation.** In most cases of interest, the objective $\mathcal{L}$ is differentiable and is usually accessed through a first-order oracle returning values of the vector field $V(\theta, \phi) = (\nabla_\theta \mathcal{L}(\theta, \phi), -\nabla_\phi \mathcal{L}(\theta, \phi))$. As usual for gradient-based methods, we will frequently (though not always) assume that $V$ is *Lipschitz continuous*:

**Assumption 1.** The field $V$ is $\beta$-Lipschitz continuous i.e., for all $x, x' \in \mathbb{R}^d$,

$$\|V(x') - V(x)\| \leq \beta \|x' - x\|. \tag{LC}$$

The importance of the above is that (SP) is often intractable, so it is natural to examine instead the first-order stationarity conditions for $V$, i.e., the problem:

$$\text{Find } x^\star \in \mathbb{R}^d \text{ such that } V(x^\star) = 0. \tag{Opt}$$

This "vector field formulation" is the unconstrained case of what is known in the literature as a *variational inequality* (VI) problem – see e.g., Facchinei & Pang [5] for a comprehensive introduction. In what follows, we will not need the full generality of the VI framework and we will develop our results in the context of (Opt) above; our only blanket assumption in this regard is that the set of solutions $\mathcal{X}^\star$ of (Opt) is nonempty.

**Feedback assumptions** Throughout the sequel, we will assume that the optimizer can access $V$ via a *stochastic first-order oracle* (SFO). This means that at every stage $t$ of an iterative algorithm, the optimizer can call this black-box mechanism at a point $X_t \in \mathbb{R}^d$ to get a feedback of the form $\hat{V}_t = V(X_t) + Z_t$ where $Z_t \in \mathbb{R}^d$ is an additive noise variable. Our bare-bones assumptions for this oracle will then be as follows:

**Assumption 2.** The noise term $Z_t$ of SFO satisfies

a) *Zero-mean:* $\qquad \mathbb{E}[Z_t \mid \mathcal{F}_t] = 0.$ \hfill (1a)

b) *Variance control:* $\quad \mathbb{E}[\|Z_t\|^2 \mid \mathcal{F}_t] \leq (\sigma + \kappa\|X_t - x^\star\|)^2 \text{ for all } x^\star \in \mathcal{X}^\star.$ \hfill (1b)

where $\sigma, \kappa \geq 0$ and $\mathcal{F}_t$ denotes the history (natural filtration) of $X_t$.

It is important to note that in (1b), $\sigma$ and $\kappa$ play different roles. When $\kappa = 0$, the condition corresponds to the classic bounded variance assumption on the noise. At the other end of the spectrum, $\sigma = 0$ implies that the noise vanish on the solution set. This kind of condition has been popularized recently in the machine learning community under the name of interpolation [34]. In the most general case, we have both $\sigma > 0$ and $\kappa > 0$; then condition (1b) allows the variance of the noise to exhibit quadratic growth with respect to the distance to the solution set. For example, for a stochastic oracle of the form $\hat{V}_t = \hat{V}(\xi, X_t)$ where $\xi$ is a random variable and $\hat{V}$ is a Carathéodory function,[2] this is trivially satisfied if $\hat{V}(\xi, \cdot)$ is Lipschitz and the variance of the noise is bounded on $\mathcal{X}^\star$. Therefore, Assumption 2 is fairly weak and verified by most relevant problems.

## 3 The extragradient method and its limitations

As discussed earlier, the go-to method for saddle-point problems and variational inequalities is the *extragradient* (EG) algorithm of Korpelevich [13] and its variants. Formally, in the general setting of the previous section, the EG algorithm can be stated recursively as:

$$X_{t+\frac{1}{2}} = X_t - \gamma_t \hat{V}_t , \qquad X_{t+1} = X_t - \gamma_t \hat{V}_{t+\frac{1}{2}} \tag{EG}$$

where $\gamma_t > 0$ is a variable stepsize sequence. Heuristically, the basic idea of the method is as follows: starting from a *base* state $X_t$, the algorithm first performs a look-ahead step to generate an intermediate – or *leading* – state $X_{t+\frac{1}{2}}$; subsequently, the oracle is called at $X_{t+\frac{1}{2}}$, and the method proceeds to a new state $X_{t+1}$ by taking a step from the *base* state $X_t$. Hence, the generation of the leading state can be seen as an *exploration* step while the second part is the bona fide *update* step.

One of the reasons for the widespread popularity of (EG) is that it achieves convergence in all monotone problems, without suffering from the non-convergence phenomena (limit cycles or otherwise) that plague vanilla one-step gradient algorithms [5]. However, this guarantee requires the method to be run with deterministic, *perfect* oracle feedback (i.e., $Z_t = 0$ for all $t$); if the method is run with genuinely stochastic feedback, the situation is considerably more complicated.

To understand the issues involved, it will be convenient to consider the following elementary example:

$$\min_{\theta \in \mathbb{R}} \max_{\phi \in \mathbb{R}} \theta\phi. \tag{2}$$

Trivially, the vector field associated to (2) is $V(\theta, \phi) = (\phi, -\theta)$ and the problem's unique solution is $(\theta^\star, \phi^\star) = (0, 0)$. Given the problem's simple structure, one would expect that (EG) should be easily capable of reaching a solution; however, as we show below, this is not the case.

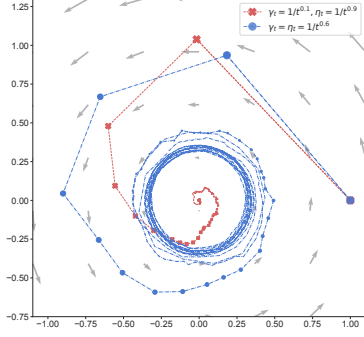

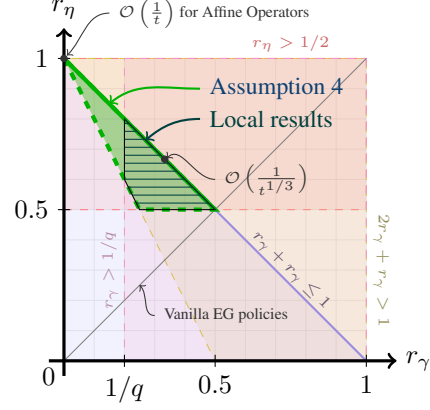

**Figure 1:** Behavior of (EG) and (DSEG) on Problem (2) with Gaussian oracle noise. Even with a vanishing, square-summable stepsize $\gamma_t = 1/t^{0.6}$, the iterates of (EG) cycle; in contrast, (DSEG) with $\gamma_t = 1/t^{0.1}$ and $\eta_t = 1/t^{0.9}$ converges.

**Figure 2:** The stepsize exponents allowed by Assumption 4 for convergence (shaded green). Dashed lines are strict frontiers. Note that vanilla EG (the separatrix $r_\gamma = r_\eta$) passes just outside of this region, explaining the method's failure.

**Proposition 1.** *Suppose that* (EG) *is run on the problem* (2) *with oracle feedback* $\hat{V}_t = V(\theta_t, \phi_t) + (\xi_t, 0)$ *for some zero-mean random variable* $\xi_t$ *with variance* $\sigma^2 > 0$. *We then have* $\liminf_{t \to \infty} \mathbb{E}[\theta_t^2 + \phi_t^2] > 0$, *i.e., the iterates of* (EG) *remain on average a positive distance away from* 0.

Importantly, Proposition 1 places *no* restrictions on the algorithm's stepsize sequence and the variance of the noise could be arbitrarily small. Relegating the details to the appendix, the key to showing this result is the recursion

$$\mathbb{E}[\theta_{t+1}^2 + \phi_{t+1}^2] = (1 - \gamma_t^2 + \gamma_t^4)\,\mathbb{E}[\theta_t^2 + \phi_t^2] + (1 + \gamma_t^2)\gamma_t^2\,\sigma^2.$$

from which it follows that $\liminf_t \mathbb{E}[\theta_t^2 + \phi_t^2] > 0$. In turn, this implies that the iterates of (EG) remain on average a positive distance away from the origin. This behavior is illustrated clearly in Fig. 1 which shows a typical non-convergent trajectory of (EG) in the planar problem (2).

## 4 Extragradient with stepsize scaling

At a high level, Proposition 1 suggests that the benefit of the exploration step is negated by the noise as the iterates of (EG) get closer to the problem's solution set. To rectify this issue, we will consider a more flexible, *double stepsize extragradient* (DSEG) method of the form

$$X_{t+\frac{1}{2}} = X_t - \gamma_t \hat{V}_t, \qquad X_{t+1} = X_t - \eta_t \hat{V}_{t+\frac{1}{2}}, \tag{DSEG}$$

with $\gamma_t \geq \eta_t > 0$. The key idea in (DSEG) is that the scaling of the method's stepsize parameters affords us an extra degree of freedom which can be tuned to order. In particular, motivated by the failure of (EG) described in the previous section, we will take a stepsize scaling schedule in which the exploration step evolves at a more aggressive time-scale compared to the update step. In so doing, the method will keep exploring (possibly with a near-constant stepsize) while maintaining a cautious update policy that does not blindly react to the observed oracle signals.

For illustration and comparison, we plot in Fig. 1 an instance of this method with a fairly aggressive exploration schedule and a respectively conservative update policy. In contrast to (EG), the iterates of (DSEG) now converge to a solution. We encode this as a positive counterpart to Proposition 1 below:

**Proposition 1′.** *Suppose that* (DSEG) *is run on the problem* (2) *with oracle feedback* $\hat{V}_t = V(\theta_t, \phi_t) + (\xi_t, 0)$ *for some zero-mean random variable* $\xi_t$ *with variance* $\sigma^2 > 0$. *If the method's stepsize policies are of the form* $\gamma_t = 1/t^{r_\gamma}$ *and* $\eta_t = 1/t^{r_\eta}$ *for some* $r_\eta > r_\gamma \geq 0$ *with* $r_\gamma + r_\eta \leq 1$, *we have* $\lim_{t \to \infty} \mathbb{E}[\theta_t^2 + \phi_t^2] \to 0$.

From an analytic viewpoint, what distinguishes (EG) from (DSEG) is the following refined bound:

**Lemma 1.** *Under Assumptions 1 and 2, for all* $t = 1, 2, \ldots$ *and all* $x^\star \in \mathcal{X}^\star$, *it holds*

$$\mathbb{E}[\|X_{t+1} - x^\star\|^2 \mid \mathcal{F}_t] \leq (1 + C_t\,\kappa^2)\|X_t - x^\star\|^2 - 2\eta_t\,\mathbb{E}[\langle V(X_{t+\frac{1}{2}}), X_{t+\frac{1}{2}} - x^\star\rangle \mid \mathcal{F}_t]$$
$$- \gamma_t\eta_t(1 - \gamma_t^2\beta^2 - 8\gamma_t\eta_t\,\kappa^2)\|V(X_t)\|^2 + C_t\sigma^2, \tag{3}$$

*with constant $C_t = 4\gamma_t^2 \eta_t \beta + 2\gamma_t^3 \eta_t \beta^2 + 4\eta_t^2 + 16\gamma_t^2 \eta_t^2 \kappa^2$.*

The proof of Lemma 1, which we defer to the supplement, relies on a careful analysis of the update between successive iterates to separate the deterministic and the stochastic effects. Analyzing the bound of Lemma 1 term-by-term gives a clear picture of how an aggressive exploration stepsize policy can be helpful:

- The term $\gamma_t \eta_t (1 - \gamma_t^2 \beta^2 - 8\gamma_t \eta_t \kappa^2) \|V(X_t)\|^2$ provides a consistently negative contribution as long as $\sup_t \gamma_t < 1/3 \max(\beta, \kappa)$.
- The term $C_t$ is antagonistic and needs to be made as small as possible.
- The term $\mathbb{E}[\langle V(X_{t+\frac{1}{2}}), X_{t+\frac{1}{2}} - x^\star \rangle \mid \mathcal{F}_t]$ plays a lesser role since it is non-negative for variational stable problems (see upcoming Assumption 3) and is even identically zero in bilinear problems.

Therefore, to obtain convergence, one needs the coefficient $\gamma_t \eta_t$ to be as *large* as possible and, concurrently, each of the terms $\gamma_t^2 \eta_t$, $\gamma_t^3 \eta_t$, $\eta_t^2$ and $\gamma_t^2 \eta_t^2$ that appear in $C_t$ should be as *small* as possible. Formally, this would lead to the requirement $\sum_t \gamma_t \eta_t = \infty$ and $\sum_t \gamma_t^2 \eta_t + \eta_t^2 < \infty$. These conditions can be simultaneously achieved by a suitable choice of $\gamma_t$ and $\eta_t$ (cf. Proposition 1′ above), but they are *mutually exclusive* if $\gamma_t = \eta_t$. This observation is the key motivation for the scale separation between the exploration and the update mechanisms in (DSEG), and is the principal reason that (EG) fails to converge in bilinear problems.

## 5 Convergence analysis

We now proceed with our main results for the DSEG algorithm. We begin in Section 5.1 with an asymptotic convergence analysis for (DSEG); subsequently, in Section 5.2, we examine the algorithm's rate of convergence; finally, in Section 5.3, we zero in on affine problems. Given our interest in non-monotone problems, we make a clear distinction between global results (which require global assumptions) and local ones (which apply to more general problems).

### 5.1 Asymptotic convergence

**Global convergence.** Our assumption for global convergence is a variational stability condition.

**Assumption 3.** The operator $V$ satisfies $\langle V(x), x - x^\star \rangle \geq 0$ for all $x \in \mathbb{R}^d$, $x^\star \in \mathcal{X}^\star$.

Assumption 3 is verified for all monotone operators but it also encompasses a wide range of non-monotone problems; for an overview see e.g., [5, 10, 12, 16, 21] and references therein.

To leverage this assumption, we will further need the algorithm's update step to decrease sufficiently quickly relative to the corresponding exploration step. Formally (and with a fair degree of hindsight), this boils down to the following:

**Assumption 4.** The stepsizes of (DSEG) satisfy $\sum_t \gamma_t \eta_t = \infty$, $\sum_t \eta_t^2 < \infty$, and $\sum_t \gamma_t^2 \eta_t < \infty$.

Assumption 4 essentially posits that $\eta_t / \gamma_t \to 0$ as $t \to \infty$, so it reflects precisely the principle of "aggressive exploration, conservative updates". In particular, Assumption 4 rules out the choice $\gamma_t = \eta_t$ which would yield the vanilla EG algorithm, providing further evidence for the use of a double stepsize policy. A typical stepsize policy for (DSEG) is

$$\gamma_t = \frac{\gamma}{(t+b)^{r_\gamma}} \quad \text{and} \quad \eta_t = \frac{\eta}{(t+b)^{r_\eta}} \tag{4}$$

for some $\gamma, \eta, b > 0$ and exponents $r_\gamma, r_\eta \in [0, 1]$. Assumption 4 then translates as $r_\gamma + r_\eta \leq 1$, $2r_\eta > 1$, and $2r_\gamma + r_\eta > 1$ as represented in Fig. 2. With this in mind, we have the following convergence result.

**Theorem 1.** *Let Assumptions 1–4 hold and $\sup_t \gamma_t < 1/3 \max(\beta, \kappa)$, then the iterates $X_t$ of (DSEG) converge almost surely to a solution $x^\star$ of (Opt).*

As far as we are aware, this is the first result of this type for stochastic first-order methods: almost sure convergence typically requires stronger hypotheses guaranteeing that $\langle V(x), x - x^\star \rangle$ is uniformly positive when $x \notin \mathcal{X}^\star$ [12, 21]. In particular, Theorem 1 implies the almost sure convergence of the algorithm for bilinear problems like (2) where EG and standard gradient methods do not converge.

**Local convergence.** To extend Theorem 1 to fully non-monotone settings, we will consider the following local version of Assumptions 1–3 near a solution point $x^\star$:

**Assumption 1′.** The field $V$ is $\beta$-Lipschitz continuous near $x^\star$, i.e., for all $x, x'$ near $x^\star$,
$$\|V(x') - V(x)\| \leq \beta \|x' - x\|.$$

**Assumption 2′.** Let $x^\star \in \mathcal{X}^\star$ and $U$ be a neighborhood of $x^\star$. The noise term $Z_t$ of SFO satisfies

   *a) Zero-mean:*        $\mathbb{E}[Z_t \mid \mathcal{F}_t]\, \mathbb{1}_{\{X_t \in U\}} = 0.$                       (5a)

   *b) Moment control:*   $\mathbb{E}[\|Z_t\|^q \mid \mathcal{F}_t]\, \mathbb{1}_{\{X_t \in U\}} \leq (\sigma + \kappa \|X_t - x^\star\|)^q.$     (5b)

for some $q > 2$ and $\sigma, \kappa \geq 0$.

**Assumption 3′.** The operator $V$ satisfies $\langle V(x), x - x^\star \rangle \geq 0$ for all $x$ near $x^\star$.

Notice that (5b) is slightly stronger than (1b) in the sense that we now require to control the $q^{th}$ moment of the noise for some $q > 2$. Nonetheless, this condition as well as the unbiasedness assumption only need to be satisfied in a neighborhood of $x^\star$. Our next result shows that, with these modified assumptions, the DSEG algorithm converges locally to solutions with high probability:

**Theorem 2.** *Fix a tolerance level $\delta > 0$ and suppose that Assumptions 1′–3′ hold for some isolated solution $x^\star$ of (Opt). Assume further that (DSEG) is run with stepsize parameters of the form (4) with small enough $\gamma$, $\eta$ and proper choice of $r_\gamma, r_\eta$ (cf. Fig. 2). If the algorithm is not initialized too far from $x^\star$, its iterates converge to $x^\star$ with probability at least $1 - \delta$.*

The first step towards proving Theorem 2 is to show that the generated iterates stay close to $x^\star$ with arbitrarily high probability. To achieve this, one needs to control the total noise accumulating from each noisy step, a task which is made difficult by the fact that the norm of the SFO feedback can only be upper bounded recursively and thus depends on previous iterates. In the supplement, we dedicate a lemma to the study of such recursive stochastic processes, and we build our analysis on this lemma.

## 5.2 Convergence rates

**Global rate.** To study the algorithm's convergence rate, we will require the following error bound condition:

**Assumption 5.** For some $\tau > 0$ and all $x \in \mathbb{R}^d$, we have
$$\|V(x)\| \geq \tau \operatorname{dist}(x, \mathcal{X}^\star). \tag{EB}$$

This kind of error bound is standard in the literature on variational inequalities for deriving last iterate convergence rates [see e.g., 5, 18, 19, 32, 33]. In particular, Assumption 5 is satisfied by

a) *Strongly monotone operators:* here, $\tau$ is the strong monotonicity modulus.
b) *Affine operators:* for $V(x) = Mx + v$ where $M$ is a matrix of size $d \times d$ and $v$ is a $d$-dimensional vector, $\tau$ is the minimum non-zero singular value of $M$.

In this sense, Assumption 5 provides a unified umbrella for two types of problems that are typically considered to be poles apart. Our first result in this context is as follows:

**Theorem 3.** *Suppose that Assumptions 1–3 and 5 hold and assume that $\gamma_t \leq c/\beta$ with $c < 1$. Then:*

*1. If (DSEG) is run with $\gamma_t \equiv \gamma$, $\eta_t \equiv \eta$, we have:*
$$\mathbb{E}[\operatorname{dist}(X_t, \mathcal{X}^\star)^2] \leq (1 - \Delta)^{t-1} \operatorname{dist}(X_1, \mathcal{X}^\star)^2 + \frac{C}{\Delta}$$

*with constants $C = (2\gamma^2 \eta\beta + \gamma^3 \eta\beta^2 + \eta^2)\sigma^2$ and $\Delta = \gamma\eta\tau^2(1 - c^2)$.[3]*

*2. If (DSEG) is run with $\gamma_t = \gamma/(t+b)^{1-\nu}$ and $\eta_t = \eta/(t+b)^\nu$ for some $\nu \in (1/2, 1)$, we have:*
$$\mathbb{E}[\operatorname{dist}(X_t, \mathcal{X}^\star)^2] \leq \frac{C}{\Delta - r} \frac{1}{t^r} + o\left(\frac{1}{t^r}\right)$$

*where $r = \min(1 - \nu, 2\nu - 1)$ and we further assume that $\gamma\eta\tau^2(1 - c^2) > r$. In particular, the optimal rate is attained when $\nu = 2/3$, which gives $\mathbb{E}[\operatorname{dist}(X_t, \mathcal{X}^\star)^2] = \mathcal{O}(1/t^{1/3})$.*

The first part of Theorem 3 shows that, if (DSEG) is run with constant stepsizes, the initial condition is forgotten exponentially fast and the iterates converge to a neighborhood of $\mathcal{X}^\star$ (though, in line with previous results, convergence cannot be achieved in this case). To make this neighborhood small, we need to decrease both $\gamma$ and $\eta/\gamma$; this would be impossible for vanilla (EG) for which $\eta/\gamma = 1$.

The second part of Theorem 3 provides an $\mathcal{O}(1/t^{1/3})$ last-iterate convergence rate. In Section 5.3, we further improve this rate to $\mathcal{O}(1/t)$ for affine operators by exploiting their particular structure.

**Local rate.**    To study the algorithm's local rate of convergence, we will focus on solutions of (Opt) that satisfy the following Jacobian regularity condition:

**Assumption 5′.** $V$ is differentiable at $x^\star$ and its Jacobian matrix $\mathrm{Jac}_V(x^\star)$ is invertible.

The link between Assumptions 5 and 5′ is provided by the following proposition:

**Proposition 2.** *If a solution $x^\star$ satisfies Assumption 5′, it satisfies (EB) in a neighborhood of $x^\star$.*

The proof of Proposition 2 follows by performing a Taylor expansion of $V$ and invoking the minimax characterization of the singular values of a matrix; we give the details in the supplement. For our purposes, what is more important is that (EB) has now been reduced to a *pointwise* condition; under this much lighter requirement, we have:

**Theorem 4.** *Fix a tolerance level $\delta > 0$ and suppose that Assumptions 1′–3′ and 5′ hold for some isolated solution $x^\star$ of (Opt) with $q > 3$. Assume further $x^\star$ satisfies Assumption 5′ and (DSEG) is run with stepsize parameters of the form $\gamma_t = \gamma/(t+b)^{1/3}$ and $\eta_t = \eta/(t+b)^{2/3}$ with large enough $b, \eta > 0$. Then, there exist neighborhoods $U, U'$ of $x^\star$ and an event $E_U$ such that:*

*a)* $\mathbb{P}(E_U \mid X_1 \in U) \geq 1 - \delta$.

*b)* $\mathbb{P}(X_t \in U' \text{ for all } t \mid E_U) = 1$.

*c)* $\mathbb{E}[\|X_t - x^\star\|^2 \mid E_U] = \mathcal{O}(1/t^{1/3})$

*In words, if (DSEG) is not initialized too far from $x^\star$, the iterates $X_t$ remain close to $x^\star$ with probability at least $1 - \delta$ and, conditioned on this event, $X_t$ converges to $x^\star$ at a rate $\mathcal{O}(1/t^{1/3})$ in mean square error.*

Taken together, Theorems 1 and 4 show that for all monotone stochastic problems with a non-degenerate critical point, employing the suggested stepsize policy yields an asymptotic $\mathcal{O}(1/t^{1/3})$ rate. In more detail, the last point of Theorem 4 shows that, with the same kind of stepsizes as in the second part of Theorem 3, we can retrieve a $\mathcal{O}(1/t^{1/3})$ convergence rate provided that the iterates stay close to the solution. Note that this rate is not a localization of Theorem 3 because, after conditioning, *the unbiasedness of the noise is not guaranteed.* To overcome this issue, our proof draws inspiration from Hsieh et al. [9] but the use of double stepsizes requires a much more intricate analysis which is reflected in the stronger noise assumption.

## 5.3   A case study of affine operators

We terminate our analysis with a dedicated treatment of affine operators which are commonly studied as a first step to understand the training of GANs [1, 4, 7, 15, 22, 35]. The following result improves the $\mathcal{O}(1/t^{1/3})$ rate of Theorem 3 to $\mathcal{O}(1/t)$ for affine operators.

**Theorem 5.** *Let $V$ be an affine operator satisfying Assumption 3, and suppose that Assumption 2 holds. Take a constant exploration stepsize $\gamma_t \equiv \gamma \leq c/\beta$ with $c < 1$ (here $\beta$ is the largest singular value of the associated matrix). Then, the iterates $(X_t)_{t \in \mathbb{N}}$ of (DSEG) enjoy the following rates:*

*1. If the update stepsize is constant $\eta_t \equiv \eta \leq \gamma$, then:*

$$\mathbb{E}[\mathrm{dist}(X_t, \mathcal{X}^\star)^2] \leq (1 - \Delta)^{t-1} \mathrm{dist}(X_1, \mathcal{X}^\star)^2 + \frac{C}{\Delta}$$

*with $C = \eta^2(1 + c^2)\sigma^2$ and $\Delta = \gamma \eta \tau^2 (1 - c^2)$.*

*2. If the update stepsize is of the form $\eta_t = \eta/(t + b)$ for $\eta > 1/(\tau^2\gamma(1 - c^2))$ and $b > \eta/\gamma$, then:*

$$\mathbb{E}[\mathrm{dist}(X_t, \mathcal{X}^\star)^2] \leq \frac{C}{\Delta - 1}\frac{1}{t} + o\left(\frac{1}{t}\right).$$

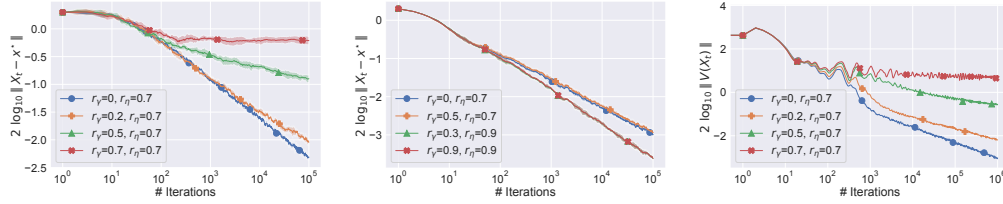

**Figure 3:** Convergence of a (DSEG) scheme in stochastic bilinear (left), strongly convex-concave (middle) and non convex-concave linear quadratic Gaussian GAN (right) problems. All curves are averaged over 10 runs with the shaded area indicating the standard deviation. The benefit of aggressive exploration is evident.

The proof of this theorem relies on the derivation of another descent lemma similar to Lemma 1 but tailored to affine operators. Note also that Assumptions 1 and 5 are automatically verified in this case.

Theorem 5 mirrors Theorem 3; however, in Part 1 of Theorem 5, the final precision is only determined by $\sigma^2$ and $\eta/\gamma$. Thus, compared to Theorem 3, there is no need to decrease $\gamma$ to obtain an arbitrarily high accuracy solution. The weaker dependence on $\gamma$ is further confirmed by Part 2, which shows a $\mathcal{O}(1/t)$ rate with $\gamma_t$ constant. As far as we are aware, this result gives the best convergence rate for stochastic affine operators compared to the literature, and it gives yet another motivation for the use of a double stepsize strategy.

## 6 Numerical experiments

This section investigates numerically the benefits of double stepsizes. We run (DSEG) with stepsize of the form (4) on three different problems: *i)* a bilinear zero-sum game, *ii)* a strongly convex-concave game and *iii)* a non convex-concave linear quadratic Gaussian GAN model [4, 25]. We examine their behavior when $r_\gamma$ and $r_\eta$ vary. The exact description of the problems and the experimental details are deferred to the supplement.

As shown in Fig. 3, for bilinear game and Gaussian GAN examples, choosing $r_\eta < r_\gamma$ turns out to be necessary for the convergence of the algorithm, and the convergence speed is positively related to the difference $r_\gamma - r_\eta$, as per our analysis. For a strongly convex-concave problem, it is known that the iterates produced by (EG) with noisy feedback achieve $\mathcal{O}(1/t)$ convergence for proper choice of $(\gamma_t)_{t\in\mathbb{N}}$ [9, 12]. Our experiment moreover reveals that when a double step-size policy is considered, the convergence speed of the algorithm seems to only depend on $(\eta_t)_{t\in\mathbb{N}}$ and using aggressive $(\gamma_t)_{t\in\mathbb{N}}$ has little influence, if any, suggesting that taking a larger exploration step may be a universal solution. Going one step further, we conduct experiments and observe similar phenomena for the generalized optimistic gradient method [24, 30] when the output vector is appropriately chosen. We refer the interested reader to the supplement for a dedicated discussion.

## 7 Conclusion

In this paper, we examined the benefits of employing a *double stepsize extragradient* method for which the exploration step is more aggressive than the update step. This additional flexibility turns out to be both necessary and sufficient for the method to achieve superior convergence properties relative to vanilla stochastic extragradient methods in a large spectrum of problems including bilinear games and some non convex-concave models.

Our results constitute a first attempt towards designing an algorithm that provably avoids cycles and similar non-convergent phenomena in a fully stochastic setting. Several interesting future directions include an extended analysis with relaxation of the variational stability assumption as well as the design of a fully adaptive and/or universal method on the basis of our results.

## Broader impact

This work does not present any foreseeable societal consequence.

## Acknowledgments

This work has been partially supported by MIAI@Grenoble Alpes, (ANR-19-P3IA-0003).

## Footnotes

[1]Let us still mention the work of Loizou et al. [17] which appeared on arxiv a few weeks after the submission of our manuscript: it proved that stochastic Hamiltonian methods applied to (sufficiently) bilinear games ensures also a $\mathcal{O}(1/t)$ convergence rate. Nonetheless, Hamiltoninan gradient descent is not guaranteed to converge to a solution in monotone games and in general when it converges, it may converge to an unstable stationary point.

[2]That is, $\hat{V}(\xi, \cdot)$ is continuous for almost all $\xi$ and $\hat{V}(\cdot, x)$ is measurable for all $x$.

[3] For better readability, these constants are stated for the case $\kappa = 0$. On the other hand, if $\sigma = 0$ (and $\kappa \geq 0$), a geometric convergence can be proved. The same arguments apply to Theorem 5.

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
