[Supplementary Material]

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

# A   Additional related work

The first analysis of *extragradient* (EG) with stochastic feedback traces back to the work of Juditsky et al. [14], where a $\mathcal{O}(1/\sqrt{t})$ ergodic convergence was shown for monotone problems, and this rate is known to be optimal without further assumptions [30].[4] Since then, a large number of works have been dedicated to studying the convergence behavior of stochastic EG-type algorithms, either for better understanding of the algorithm itself or in the hope of finding a better way to incorporate EG with stochasticity.

Almost sure convergence of stochastic EG was first investigated in Kannan & Shanbhag [15]. In the said paper, almost convergence was shown for *pseudomonotone plus* operators and by additionally assuming that the map is *strongly pseudomonotone* or *monotone and weak-sharp*, the authors managed to prove a $\mathcal{O}(1/t)$ convergence of the iterate produced by the algorithm. In [24], the pseudo-monotonicity-plus assumption is relaxed to show that stochastic EG still enjoys last-iterate convergence in *strict coherent* problems. Nonetheless, these results fail to justify the use of EG for stochastic monotone problems, as illustrated in Section 3. Therefore, to improve the convergence behavior of EG in stochastic problems, several modifications to the original stochastic EG have been proposed [2, 12, 26]. In addition to the ones discussed in Section 1, Mishchenko et al. [26] advocated a repeated sampling strategy and illustrated numerically its better performance when applied to GAN training. They also showed that their proposed algorithm retain the same convergence guarantee as traditional stochastic EG.

In order to reduce the overall computational cost, another line of research aims at designing optimization methods that solve variational problems with a single oracle call per iteration (instead of the two in EG). Algorithms of this family include for example *optimistic gradient* (OG) [5] and *extragradient with extrapolation from the past* (PEG) [9, 36]. See Hsieh et al. [11] for a recent overview and corresponding treatment in the stochastic setting. Very recently, the convergence of stochastic OG are further improved in two different ways. In [7], the authors introduced a multistage version of OG for stochastic strongly monotone problems to optimize the dependence of convergence speed on initial error and noise characteristics. On the other hand, inspired by the success of adaptive methods in deep learning, Liu et al. [19] designed an adaptive variant of OG and showed that it enjoyed an adaptive complexity that varies according to the growth rate of the cumulative stochastic gradient. To complete the list, also in the goal of reducing overall computation though under a quite different perspective, Jelassi et al. [13] analyzed a randomized version of stochastic EG in multiplayer game to make the extrapolation step amenable to massive multiplayer settings.

# B   Generalized optimistic gradient

Considering the similarity between EG and its single-call variants, we believe our analysis on (DSEG) also suggests essential modifications in terms of stepsizes that should be carried out for these algorithms in the face of stochasticity. As an example, we investigate the OG method of Daskalakis et al. [5], and find out that some surprising conclusions can be drawn after applying the double stepsize rule. The generalized OG recursion is commonly stated as follows [27, 38]:

$$X_{t+1} = X_t - \eta_t \hat{V}_t - \gamma_t(\hat{V}_t - \hat{V}_{t-1}) \tag{OG}$$

where $\gamma_t$ is sometimes called the *optimism* rate. Similarly to our conclusions, it has been empirically observed that taking large optimism rate often yields better convergence in stochastic problems [33].

Hsieh et al. [11] pointed out that OG is equivalent to the modified Arrow-Hurwitz method introduced by Popov [36] and also referred to as PEG by Gidel et al. [9]. Using a double stepsize policy, PEG becomes:

$$X_{t+\frac{1}{2}} = X_t - \gamma_t \hat{V}_{t-\frac{1}{2}}, \quad X_{t+1} = X_t - \eta_t \hat{V}_{t+\frac{1}{2}}. \tag{DSPEG}$$

Hence, leading states can be recursively written as

$$X_{t+\frac{1}{2}} = X_{t-\frac{1}{2}} - \eta_{t-\frac{3}{2}} \hat{V}_{t-\frac{1}{2}} - \gamma_{t-\frac{1}{2}} \hat{V}_{t-\frac{1}{2}} + \gamma_{t-\frac{3}{2}} \hat{V}_{t-\frac{3}{2}}.$$

We thereby see that (OG) and (DSPEG) are almost equivalent and they mostly differ in the choice of vectors that the method outputs at the end: OG suggests outputting $X_t$ while PEG instead looks

**Figure 4:** Convergence of (DSEG) (top) and (OG) (bottom) schemes in stochastic bilinear (left), strongly convex-concave (middle) and non convex-concave covariance matrix learning (right) problems. In the second row the dashed lines and the solid lines depict respectively the results for optimistic iterates and residual iterates. We observe clearly the benefit of *(i)* aggressive exploration and *(ii)* using residual iterates in generalized OG methods. All curves are averaged over 10 runs with the shaded area indicating the standard deviation.

at $X_t + \gamma_{t-1}\hat{V}_{t-1}$. This nuance turns out to be of importance when generalized OG is applied to stochastic problems. By analogy with our analysis for (DSEG), we reasonably conjecture that taking $\eta_t < \gamma_t$ guarantees the convergence of $X_t + \gamma_{t-1}\hat{V}_{t-1}$, and this may occur even if $\gamma_t$ is set to constant. Nonetheless, this also implies that if the noise is not vanishing at the solution, $X_t$, which corresponds to the exploration state in PEG, might exhibit much slower convergence or even not converge at all.

To summarize, when running (OG) for stochastic problems, we should look at the *residual iterate* $X_t + \gamma_{t-1}\hat{V}_{t-1}$ instead of the *optimistic iterate* $X_t$. Interestingly, this conclusion is consistent with the ODE analysis of OG by Ryu et al. [38], and explains some experimental results of said work. Furthermore, taking an aggressive exploration step $\gamma_t$ and a more conservative update step $\eta_t$ may be very beneficial both in theory (for the last iterate convergence and rate) and in practice as confirmed by our experiments just below.

## C  Experimental details and additional experiments

We provide here a detailed explanation of the problems that we consider in our experiments and elucidate the used parameters. Additional experimental results are also presented.

**Bilinear zero-sum games.**   The bilinear zero-sum game takes the form

$$\mathcal{L}(\theta, \phi) = \theta^\top C \phi$$

where $C$ is a $50 \times 50$ invertible matrix in our experiment; in that case, $(\theta^\star, \phi^\star) = (0, 0)$ is the only equilibrium point. We simulate the stochastic oracle by adding a Gaussian noise $Z \sim \mathcal{N}(0, \sigma I)$ with $\sigma = 0.5$ to the vector field.

**Strongly convex-concave game.**   To understand the effect of aggressive exploration in strongly convex-concave problems, we inspect the following example

$$\mathcal{L}(\theta, \phi) = \left(\theta^\top A_2 \theta\right)^2 + 2\theta^\top A_1 \theta + 4\theta^\top C \phi - 2\phi^\top B_1 \phi - \left(\phi^\top B_2 \phi\right)^2,$$

where $A_1$, $A_2$, $B_1$, $B_2$ are $50 \times 50$ positive definite matrices so $(\theta^\star, \phi^\star) = (0, 0)$ is again the only solution of the problem. We take the same noise distribution to construct the stochastic oracle.

**Linear Quadratic Gaussian GAN.**   Finally, to examine the convergence of (DSEG) in stochastic non convex-concave problems, we consider the following problem from Daskalakis et al. [5] and Nagarajan & Kolter [28]:

$$\mathcal{L}(Y, W) = \mathbb{E}_{x \sim \mathcal{N}(0, \Sigma)}[x^\top W x] - \mathbb{E}_{z \sim \mathcal{N}(0, I)}[z^\top Y^\top W Y z].$$

|  | Double stepsize extragradient (DSEG) | | | Generalized optimistic gradient (OG) | | |
|---|---|---|---|---|---|---|
|  | $\gamma_1$ | $\eta_1$ | $b$ | $\gamma_1$ | $\eta_1$ | $b$ |
| Bilinear | 1 | 0.1 | 19 | 0.5 | 0.05 | 19 |
| Strongly convex-concave | 0.1 | 0.05 | 19 | 0.1 | 0.05 | 19 |
| Gaussian GAN | 0.5 | 0.05 | 49 | 0.05 | 0.025 | 99 |

**Table 2:** The stepsize parameters for (DSEG) and (OG) in the experiments.

This saddle-point problem corresponds to the WGAN formulation without clipping when data are sampled from a normal distribution with covariance matrix $\Sigma$, i.e., $x \sim \mathcal{N}(0, \Sigma)$, and the generator and the discriminator are respectively defined by $G(z) = Yz$, $D(x) = x^\top W x$. The stochasticity is induced by the sampling of $x$ and $z$. For the experiments we take a mini-batch of size 128 and $x$ and $z$ of dimension 10. As the game may possess multiple equilibria, the squared norm of $V$ is traced as the convergence measure.

**Results for (DSEG) and (OG).** Following the discussion of Appendix B, we complement the illustration of our method (DSEG) by a comparison with (OG) with properly chosen outputs. In the experiments, both (DSEG) and (OG) are run with stepsize of the form (4) with various $r_\gamma$ and $r_\eta$. In order to start with the same value for different exponents, we fix $b, \gamma_1$, and $\eta_1$ as indicated in Table 2, from which we deduce $\gamma = \gamma_1(1 + b)^{r_\gamma}$ and $\eta = \eta_1(1 + b)^{r_\eta}$.

As shown in Fig. 4, for bilinear game and Gaussian GAN examples, the convergence speed of (DSEG) is positively related to the difference $r_\gamma - r_\eta$, as per our analysis. For the strongly convex-concave problem, the vanilla (EG) already achieves $\mathcal{O}(1/t)$ convergence, and the plot shows that using aggressive $(\gamma_t)_{t \in \mathbb{N}}$ has little influence on it.

Regarding (OG) with the residual iterates, the algorithm has roughly the same convergence behavior as for (DSEG). In contrast, the optimistic iterates tend to converge much slower. In particular, choosing a constant exploration step gives the fastest convergence of the residual iterate though the optimistic iterate does not converge, in line with our discussion in Appendix B.

**Additional discussions for bilinear games.** Few algorithms provably converge in stochastic bilinear games, and among them there are stochastic Hamiltonian gradient descent (SHGD) [20] and gradient descent with anchoring [38]. In Fig. 5 we illustrate the convergences of DSEG and these two algorithms for the stochastic bilinear saddle-point example. For (DSEG) we adopt the optimal stepsize schedule as described in Theorem 5-2. The leading stepsize is set to constant $\gamma_t \equiv 1$ and the update stepsize is $\eta_t = \eta/(t + b)$ with $\eta = 2$ and $b = 19$. The same $(\eta_t)_{t \in \mathbb{N}}$ is also used as the stepsize of SHGD, in accordance with the decreasing stepsize strategy presented in [20]. As for the anchored gradient methods, its update is written as

$$X_{t+1} = X_t - \frac{1 - r}{t^r} + \frac{(1 - r)\gamma}{t^\nu}(X_1 - X_t),$$

and it is proved to converge in all stochastic monotone problems for $\gamma > 0$ and $r, \nu \in (1/2, 1)$. Since no explicit rate is proven for this algorithm when stochastic gradients are used, we run hyperparameter optimization to search for the best $\gamma, r$ and $\nu$, and end up with $\gamma = 1, r = 0.7, \nu = 0.9$.

Fig. 5 confirms that asymptotically both DSEG and SHGD converge in $\mathcal{O}(1/t)$ as predicted by the theory. SHGD converges slightly faster than DSEG for the first few iterations as it circumvents the rotational dynamics by directly performing stochastic gradient descent on $\|V(\cdot)\|^2$, which turns out to be a positive definite quadratic form when $V$ is linear. This however comes at the cost of the use of second-order information. In fact, SHGD requires access to an unbiased estimator of $\mathrm{Jac}_V^\top V$ at every iteration. Finally, anchoring converges much slower compared to these two methods. Without further theoretical investigation we do not know if this kind of algorithms can achieve the same $\mathcal{O}(1/t)$ convergence rate in this problem.

**Figure 5:** Comparison of DSEG, stochastic Hamiltonian gradient descent and anchored gradient in the stochastic bilinear example. All curves are averaged over 10 runs with the shaded area indicating the standard deviation.

# D  Technical lemmas

In this section we recall several important lemmas that are frequently used in the analysis of stochastic iterative methods. The first three lemmas on numerical sequences are useful for deriving convergence rates of the algorithms. See e.g., Polyak [35] for an abundance of results of this type.

**Lemma D.1.** *Let $(a_t)_{t \in \mathbb{N}}$ be a sequence of real numbers such that for all $t$,*

$$a_{t+1} \leq (1-q)a_t + q',$$

*where $1 > q > 0$ and $q' > 0$. Then,*

$$a_t \leq (1-q)^{t-1}a_1 + \frac{q'}{q}.$$

The above lemma comes into play when an algorithm is run with constant stepsize sequences, whereas we resort to the following two lemmas in case of decreasing stepsize sequences of the form (4).

**Lemma D.2** (Chung [4, Lemma 1]). *Let $(a_t)_{t \in \mathbb{N}}$ be a sequence of real numbers and $b \in \mathbb{N}$ such that for all $t$,*

$$a_{t+1} \leq \left(1 - \frac{q}{t+b}\right)a_t + \frac{q'}{(t+b)^{r+1}},$$

*where $q > r > 0$ and $q' > 0$. Then,*

$$a_t \leq \frac{q'}{q-r}\frac{1}{t^r} + o\left(\frac{1}{t^r}\right).$$

**Lemma D.3** (Chung [4, Lemma 4]). *Let $(a_t)_{t \in \mathbb{N}}$ be a sequence of real numbers and $b \in \mathbb{N}$ such that for all $t$,*

$$a_{t+1} \leq \left(1 - \frac{q}{(t+b)^\nu}\right)a_t + \frac{q'}{(t+b)^{r+\nu}},$$

*where $1 > \nu > 0$ and $r, q, q' > 0$. Then,*

$$a_t = \mathcal{O}\left(\frac{1}{t^r}\right).$$

To establish almost sure convergence of the iterates, we rely on the Robbins–Siegmund theorem which apply to non-negative almost-supermatingales.

**Lemma D.4** (Robbins & Siegmund [37]). *Consider a filtration $(\mathcal{F}_t)_{t \in \mathbb{N}}$ and four non-negative $(\mathcal{F}_t)_{t \in \mathbb{N}}$-adapted processes $(U_t)_{t \in \mathbb{N}}$, $(\lambda_t)_{t \in \mathbb{N}}$, $(\chi_t)_{t \in \mathbb{N}}$, $(\zeta_t)_{t \in \mathbb{N}}$ such that $\sum_t \lambda_t < \infty$ and $\sum_t \chi_t < \infty$ with probability one and $\forall t \in \mathbb{N}$,*

$$\mathbb{E}[U_{t+1} \mid \mathcal{F}_t] \leq (1 + \lambda_t)U_t + \chi_t - \zeta_t. \tag{D.1}$$

*Then $(U_t)_{t \in \mathbb{N}}$ converges almost surely to a random variable $U_\infty$ and $\sum_t \zeta_t < \infty$ almost surely.*

# E  Proofs for global convergence results

We then start with the proofs of the global results to highlight the effect of double stepsize, before tackling the more challenging local convergence analysis.

## E.1  Proof of Proposition 1: failure of stochastic extragradient

**Proposition 1.** *Suppose that (EG) is run on the problem (2) with oracle feedback $\hat{V}_t = V(\theta_t, \phi_t) + (\xi_t, 0)$ for some zero-mean random variable $\xi_t$ with variance $\sigma^2 > 0$. We then have $\liminf_{t \to \infty} \mathbb{E}[\theta_t^2 + \phi_t^2] > 0$, i.e., the iterates of (EG) remain on average a positive distance away from 0.*

*Proof.* We write the updates of the algorithm

$$\begin{cases} \theta_{t+\frac{1}{2}} = \theta_t - \gamma_t\phi_t - \gamma_t\xi_t \\ \phi_{t+\frac{1}{2}} = \phi_t + \gamma_t\theta_t \end{cases} \qquad \begin{cases} \theta_{t+1} = \theta_t - \gamma_t\phi_t - \gamma_t^2\theta_t - \gamma_t\xi_{t+\frac{1}{2}} \\ \phi_{t+1} = \phi_t + \gamma_t\theta_t - \gamma_t^2\phi_t - \gamma_t^2\xi_t \end{cases}$$

Therefore

$$\theta_{t+1}^2 + \phi_{t+1}^2 = (1 - \gamma_t^2 + \gamma_t^4)(\theta_t^2 + \phi_t^2) + \gamma_t^2\xi_{t+\frac{1}{2}}^2 + \gamma_t^4\xi_t^2$$
$$- 2\gamma_t\xi_{t+\frac{1}{2}}((1 - \gamma_t^2)\theta_t - \gamma_t\phi_t) - 2\gamma_t^2\xi_t((1 - \gamma_t^2)\phi_t + \gamma_t\theta_t).$$

Taking expectation leads to

$$\mathbb{E}[\theta_{t+1}^2 + \phi_{t+1}^2] = (1 - \gamma_t^2 + \gamma_t^4)\,\mathbb{E}[\theta_t^2 + \phi_t^2] + (\gamma_t^2 + \gamma_t^4)\sigma^2.$$

For sake of simplicity, let us denote $a_t = \mathbb{E}[\theta_t^2 + \phi_t^2]$. We consider two scenarios:

*Case 1:* $\gamma_t^2 \geq 1$.  We have $1 - \gamma_t^2 + \gamma_t^4 \geq 1$ and consequently $a_{t+1} \geq a_t$.

*Case 2:* $\gamma_t^2 < 1$.  Notice that

$$a_{t+1} - \frac{(1 + \gamma_t^2)\sigma^2}{1 - \gamma_t^2} = (1 - \gamma_t^2 + \gamma_t^4)\left(a_t - \frac{(1 + \gamma_t^2)\sigma^2}{1 - \gamma_t^2}\right).$$

We then set $\nu_t = (1 + \gamma_t^2)/(1 - \gamma_t^2)$. Since $1 - \gamma_t^2 + \gamma_t^4 < 1$, $a_{t+1}$ gets closer to $\nu_t\sigma^2$ than $a_t$. In particular, if $a_t < \nu_t\sigma^2$, we have $a_t < a_{t+1} < \nu_t\sigma^2$; otherwise, $a_t \geq a_{t+1} \geq \nu_t\sigma^2$. As $\nu_t \geq 1$, the above implies $a_{t+1} \geq \min(a_t, \nu_t\sigma^2) \geq \min(a_t, \sigma^2)$.

To conclude, in the two cases we have $a_{t+1} \geq \min(a_t, \sigma^2)$, showing $\liminf_{t\to\infty} \mathbb{E}[\theta_t^2 + \phi_t^2] > 0$.

**A remedy with double stepsize extragradient.**  With different stepsizes, the updates of the algorithm write

$$\begin{cases} \theta_{t+\frac{1}{2}} = \theta_t - \gamma_t\phi_t - \gamma_t\xi_t \\ \phi_{t+\frac{1}{2}} = \phi_t + \gamma_t\theta_t \end{cases} \qquad \begin{cases} \theta_{t+1} = \theta_t - \eta_t\phi_t - \gamma_t\eta_t\theta_t - \eta_t\xi_{t+\frac{1}{2}} \\ \phi_{t+1} = \phi_t + \eta_t\theta_t - \gamma_t\eta_t\phi_t - \gamma_t\eta_t\xi_t \end{cases}$$

This now leads to

$$\mathbb{E}[\theta_{t+1}^2 + \phi_{t+1}^2] = ((1 - \gamma_t\eta_t)^2 + \eta_t^2)\,\mathbb{E}[\theta_t^2 + \phi_t^2] + (\eta_t^2 + \gamma_t^2\eta_t^2)\sigma^2$$
$$= (1 - 2\gamma_t\eta_t + \eta_t^2 + \gamma_t^2\eta_t^2)\,\mathbb{E}[\theta_t^2 + \phi_t^2] + (\eta_t^2 + \gamma_t^2\eta_t^2)\sigma^2.$$

Taking $\gamma_t = \frac{1}{t^{r_\gamma}}$ and $\eta_t = \frac{1}{t^{r_\eta}}$, we get

$$\mathbb{E}[\theta_{t+1}^2 + \phi_{t+1}^2] = \left(1 - \frac{2}{t^{(r_\gamma + r_\eta)}} + \frac{1}{t^{2r_\eta}} + \frac{1}{t^{2(r_\gamma + r_\eta)}}\right)\mathbb{E}[\theta_t^2 + \phi_t^2] + \left(\frac{1}{t^{2r_\eta}} + \frac{1}{t^{2(r_\gamma + r_\eta)}}\right)\sigma^2$$
$$\leq \left(1 - \frac{1.5}{t^{(r_\gamma + r_\eta)}}\right)\mathbb{E}[\theta_t^2 + \phi_t^2] + \frac{2\sigma^2}{t^{2r_\eta}}$$
$$= \mathcal{O}\left(\frac{1}{t^{(r_\eta - r_\gamma)}}\right)$$

where the inequality comes from $1 - 2/t^{(r_\gamma + r_\eta)} + 1/t^{2r_\eta} + 1/t^{2(r_\gamma + r_\eta)} \leq 1 - 1.5/t^{(r_\gamma + r_\eta)}$ for large enough $t$ and the last part is an application of either Lemma D.2 or Lemma D.3 with $q = 1.5 > r = r_\eta - r_\gamma > 0$ (starting at large enough $t$).

Hence, $\mathbb{E}[\theta_t^2 + \phi_t^2] \to 0$, i.e. we can find a double stepsize choice, with an aggressive extrapolation step and a conservative update step ($r_\gamma < r_\eta$) such that $(\theta_t, \phi_t) \to (0, 0)$ in mean squared error. $\qquad\square$

## E.2  Proof of Lemma 1

**Lemma 1.** *Under Assumptions 1 and 2, for all $t = 1, 2, \ldots$ and all $x^\star \in \mathcal{X}^\star$, it holds*

$$\mathbb{E}[\|X_{t+1} - x^\star\|^2 \mid \mathcal{F}_t] \leq (1 + C_t\,\kappa^2)\|X_t - x^\star\|^2 - 2\eta_t\,\mathbb{E}[\langle V(X_{t+\frac{1}{2}}), X_{t+\frac{1}{2}} - x^\star\rangle \mid \mathcal{F}_t]$$
$$- \gamma_t\eta_t(1 - \gamma_t^2\beta^2 - 8\gamma_t\eta_t\,\kappa^2)\|V(X_t)\|^2 + C_t\sigma^2, \tag{3}$$

*with constant $C_t = 4\gamma_t^2\eta_t\beta + 2\gamma_t^3\eta_t\beta^2 + 4\eta_t^2 + 16\gamma_t^2\eta_t^2\,\kappa^2$.*

*Proof.* Let us denote by $\mathbb{E}_t[\cdot] = \mathbb{E}[\cdot \mid \mathcal{F}_t]$ the conditional expectation with respect to the filtration up to time $t$ and $\tilde{X}_{t+\frac{1}{2}} = X_t - \gamma_t V(X_t)$ the leading state that is generated with deterministic update so that $X_{t+\frac{1}{2}} = \tilde{X}_{t+\frac{1}{2}} - \gamma_t Z_t$. We develop

$$
\begin{aligned}
\|X_{t+1} - x^\star\|^2 &= \|X_t - \eta_t \hat{V}_{t+\frac{1}{2}} - x^\star\|^2 \\
&= \|X_t - x^\star\|^2 - 2\eta_t \langle \hat{V}_{t+\frac{1}{2}}, X_t - x^\star \rangle + \eta_t^2 \|\hat{V}_{t+\frac{1}{2}}\|^2 \\
&= \|X_t - x^\star\|^2 - 2\eta_t \langle \hat{V}_{t+\frac{1}{2}}, \tilde{X}_{t+\frac{1}{2}} - x^\star \rangle - 2\gamma_t \eta_t \langle \hat{V}_{t+\frac{1}{2}}, V(X_t) \rangle + \eta_t^2 \|\hat{V}_{t+\frac{1}{2}}\|^2.
\end{aligned}
$$
(E.1)

We would then like to bound the different terms appearing on the right-hand side (RHS) of the equality. With the zero-mean assumption (1a), conditioning on $\mathcal{F}_t$ leads to

$$
\begin{aligned}
\mathbb{E}_t[\langle \hat{V}_{t+\frac{1}{2}}, \tilde{X}_{t+\frac{1}{2}} - x^\star \rangle] &= \mathbb{E}_t[\langle V(X_{t+\frac{1}{2}}), \tilde{X}_{t+\frac{1}{2}} - x^\star \rangle] \\
&= \mathbb{E}_t[\langle V(X_{t+\frac{1}{2}}), \tilde{X}_{t+\frac{1}{2}} - \gamma_t Z_t - x^\star \rangle] + \mathbb{E}_t[\langle V(X_{t+\frac{1}{2}}), \gamma_t Z_t \rangle] \\
&= \mathbb{E}_t[\langle V(X_{t+\frac{1}{2}}), X_{t+\frac{1}{2}} - x^\star \rangle] + \gamma_t \, \mathbb{E}_t[\langle V(X_{t+\frac{1}{2}}) - V(\tilde{X}_{t+\frac{1}{2}}), Z_t \rangle],
\end{aligned}
$$
(E.2)

where in the last line we use the fact that $V(\tilde{X}_{t+\frac{1}{2}})$ is $\mathcal{F}_t$-measurable so

$$
\mathbb{E}_t[\langle V(\tilde{X}_{t+\frac{1}{2}}), Z_t \rangle] = \langle V(\tilde{X}_{t+\frac{1}{2}}), \mathbb{E}_t[Z_t] \rangle = 0.
$$

By Lipschitz continuity of $V$

$$
-\langle V(X_{t+\frac{1}{2}}) - V(\tilde{X}_{t+\frac{1}{2}}), Z_t \rangle \le \|V(X_{t+\frac{1}{2}}) - V(\tilde{X}_{t+\frac{1}{2}})\| \|Z_t\| \le \gamma_t \beta \|Z_t\|^2. \quad \text{(E.3)}
$$

On the other hand, $\mathbb{E}_t[\langle \hat{V}_{t+\frac{1}{2}}, V(X_t) \rangle] = \mathbb{E}_t[\langle V(X_{t+\frac{1}{2}}), V(X_t) \rangle]$ and $\mathbb{E}_t[\|\hat{V}_{t+\frac{1}{2}}\|^2] = \mathbb{E}_t[\|V(X_{t+\frac{1}{2}})\|^2] + \mathbb{E}_t[\|Z_{t+\frac{1}{2}}\|^2]$. By $\eta_t \le \gamma_t$, Lipschitz continuity of $V$ and $X_t - X_{t+\frac{1}{2}} = \gamma_t \hat{V}_t$, we get

$$
\begin{aligned}
&- 2\gamma_t \eta_t \langle V(X_{t+\frac{1}{2}}), V(X_t) \rangle + \eta_t^2 \|V(X_{t+\frac{1}{2}})\|^2 \\
&\le -2\gamma_t \eta_t \langle V(X_{t+\frac{1}{2}}), V(X_t) \rangle + \gamma_t \eta_t \|V(X_{t+\frac{1}{2}})\|^2 \\
&= \gamma_t \eta_t (\|V(X_t) - V(X_{t+\frac{1}{2}})\|^2 - \|V(X_t)\|^2) \\
&\le \gamma_t^3 \eta_t \beta^2 \|\hat{V}_t\|^2 - \gamma_t \eta_t \|V(X_t)\|^2,
\end{aligned}
$$
(E.4)

Similar to before we may write $\mathbb{E}_t[\|\hat{V}_t\|^2] = \mathbb{E}_t[\|V(X_t)\|^2] + \mathbb{E}_t[\|Z_t\|^2]$. Therefore, combining (E.1), (E.2), (E.3), (E.4), we deduce the following

$$
\begin{aligned}
\mathbb{E}_t[\|X_{t+1} - x^\star\|^2] &\le \|X_t - x^\star\|^2 - 2\eta_t \, \mathbb{E}_t[\langle V(X_{t+\frac{1}{2}}), X_{t+\frac{1}{2}} - x^\star \rangle] - (\gamma_t \eta_t - \gamma_t^3 \eta_t \beta^2) \|V(X_t)\|^2 \\
&\quad + (2\gamma_t^2 \eta_t \beta + \gamma_t^3 \eta_t \beta^2) \, \mathbb{E}[\|Z_t\|^2] + \eta_t^2 \, \mathbb{E}[\|Z_{t+\frac{1}{2}}\|^2].
\end{aligned}
$$
(E.5)

To finish the proof, we would like to bound the noise terms. Using (1b) and Jensen's inequality (recall that $q \ge 2$), we have

$$
\mathbb{E}[\|Z_t\|^2] \le (\sigma + \kappa \|X_t - x^\star\|)^2 \le 2\sigma^2 + 2\kappa^2 \|X_t - x^\star\|^2. \quad \text{(E.6)}
$$

Similarly,

$$
\begin{aligned}
\mathbb{E}[\|Z_{t+\frac{1}{2}}\|^2] &\le 2\sigma^2 + 2\kappa^2 \|X_{t+\frac{1}{2}} - x^\star\|^2 \\
&\le 2\sigma^2 + 4\kappa^2 \|X_{t+\frac{1}{2}} - X_t\|^2 + 4\kappa^2 \|X_t - x^\star\|^2 \\
&\le 4\gamma_t^2 \kappa^2 \|\hat{V}_t\|^2 + 4\kappa^2 \|X_t - x^\star\|^2 + 2\sigma^2 \\
&\le 8\gamma_t^2 \kappa^2 \|V(X_t)\|^2 + 16\gamma_t^2 \kappa^2 \sigma^2 + 16\gamma_t^2 \kappa^4 \|X_t - x^\star\|^2 + 4\kappa^2 \|X_t - x^\star\|^2 + 2\sigma^2
\end{aligned}
$$
(E.7)

Substituting (E.7) and (E.6) in (E.5), we obtain

$$
\begin{aligned}
\mathbb{E}_t[\|X_{t+1} - x^\star\|^2] &\le (1 + 4\gamma_t^2 \eta_t \beta \kappa + 2\gamma_t^3 \eta_t \beta^2 \kappa + 4\eta_t^2 \kappa^2 + 16\gamma_t^2 \eta_t^2 \kappa^4) \|X_t - x^\star\|^2 \\
&\quad - 2\eta_t \, \mathbb{E}_t[\langle V(X_{t+\frac{1}{2}}), X_{t+\frac{1}{2}} - x^\star \rangle] \\
&\quad - (\gamma_t \eta_t - \gamma_t^3 \eta_t \beta^2 - 8\gamma_t^2 \eta_t^2 \kappa^2) \|V(X_t)\|^2 \\
&\quad + (4\gamma_t^2 \eta_t \beta + 2\gamma_t^3 \eta_t \beta^2 + 2\eta_t^2 + 16\gamma_t^2 \eta_t^2 \kappa^2) \sigma^2.
\end{aligned}
$$

We recover (3) by using $2\eta_t^2 \sigma^2 \le 4\eta_t^2 \sigma^2$. $\qquad \square$

### E.3 Proof of Theorem 1

**Theorem 1.** *Let Assumptions 1–4 hold and* $\sup_t \gamma_t < 1/3 \max(\beta, \kappa)$, *then the iterates* $X_t$ *of* (DSEG) *converge almost surely to a solution* $x^\star$ *of* (Opt).

*Proof.* The proof is divided into three key steps.

(1) *With probability* 1, $\liminf_{t\to\infty} \|V(X_t)\| = 0$. Let $x^\star \in \mathcal{X}^\star$. Using Lemma 1 and Assumption 3, we get the following

$$
\mathbb{E}[\|X_{t+1} - x^\star\|^2 \mid \mathcal{F}_t] \leq (1 + C_t\,\kappa^2)\|X_t - x^\star\|^2 - 2\eta_t\,\mathbb{E}[\langle V(X_{t+\frac{1}{2}}), X_{t+\frac{1}{2}} - x^\star\rangle \mid \mathcal{F}_t]
$$
$$
- \gamma_t\eta_t(1 - \gamma_t^2\beta^2 - 8\gamma_t\eta_t\,\kappa^2)\|V(X_t)\|^2 + C_t\sigma^2,
$$
$$
\leq (1 + C_t\,\kappa^2)\|X_t - x^\star\|^2 - \gamma_t\eta_t(1 - \gamma_t^2\beta^2 - 8\gamma_t\eta_t\,\kappa^2)\|V(X_t)\|^2 + C_t\sigma^2
$$

Since $\gamma_t < 1/3 \max(\beta, \kappa)$ and $\eta_t \leq \gamma_t$, the coefficient $\rho_t := \gamma_t\eta_t - \gamma_t^3\eta_t\beta^2 - 8\gamma_t^2\eta_t^2\,\kappa^2$ is non-negative. Recalling that $C_t = 4\gamma_t^2\eta_t\beta + 2\gamma_t^3\eta_t\beta^2 + 4\eta_t^2 + 16\gamma_t^2\eta_t^2\,\kappa^2$, from our stepsize conditions $\sum_t \eta_t^2 < \infty$, $\sum_t \gamma_t^2\eta_t < \infty$ and $(\gamma_t)_{t\in\mathbb{N}}$ being upper-bounded, it holds $\sum_t C_t < \infty$. We can therefore apply the Robbins–Siegmund theorem (Lemma D.4) to get that *(i)* $\|X_t - x^\star\|$ converges almost surely and *(ii)* $\sum_t \rho_t\|V(X_t)\|^2 < \infty$ almost surely. As the stepsize conditions also imply $\sum_t \rho_t = \infty$, using *(ii)*, we deduce immediately $\liminf_{t\to\infty}\|V(X_t)\| = 0$ almost surely.

(2) *With probability* 1, $\|X_t - x^\star\|$ *converges for all* $x^\star \in \mathcal{X}^\star$. In other words, we would like to prove the existence of an event $\mathcal{E} \subset \Omega$ satisfying $\mathbb{P}(\mathcal{E}) = 1$ and that for every realization of the event and every $x^\star \in \mathcal{X}^\star$, $\|X_t - x^\star\|$ converges. Since $\mathbb{R}^d$ is a separable metric space, $\mathcal{X}^\star$ is also separable and we can find a countable set $\mathcal{Z}$ such that $\mathcal{X}^\star = \mathrm{cl}(\mathcal{Z})$ ($\mathcal{X}^\star$ is closed by continuity of $V$). We claim that the choice $\mathcal{E} = \{\|X_t - z\| \text{ converges for all } z \in \mathcal{Z}\}$ is the good candidate.

In effect, taking an arbitrary $z$ from $\mathcal{Z}$, from *(i)* we know that

$$
\mathbb{P}(\{\|X_t - z\| \text{ converges}\}) = 1.
$$

Therefore from the countability of $\mathcal{Z}$ we have $\mathbb{P}(\mathcal{E}) = 1$. We now fix $x^\star \in \mathcal{X}^\star$. As $\mathcal{Z}$ is dense in $\mathcal{X}^\star$, there exists a sequence $(z_i)_{i\in\mathbb{N}}$ of points in $\mathcal{Z}$ such that $\lim_{i\to\infty} z_i = x^\star$. Consider a realization of $\mathcal{E}$, for every $z_i$ we have $\lim_{t\to\infty}\|X_t - z_i\| = \nu_i$ for some $\nu_i \geq 0$. The triangular inequality gives

$$
-\|z_i - x^\star\| \leq \|X_t - x^\star\| - \|X_t - z_i\| \leq \|z_i - x^\star\|
$$

for all $i, t \in \mathbb{N}$. Consequently, for all $i \in \mathbb{N}$,

$$
-\|z_i - x^\star\| \leq \liminf_{t\to\infty}\|X_t - x^\star\| - \lim_{t\to\infty}\|X_t - z_i\|
$$
$$
= \liminf_{t\to\infty}\|X_t - x^\star\| - \nu_i
$$
$$
\leq \limsup_{t\to\infty}\|X_t - x^\star\| - \nu_i
$$
$$
= \limsup_{t\to\infty}\|X_t - x^\star\| - \lim_{t\to\infty}\|X_t - z_i\| \leq \|z_i - x^\star\|.
$$

Taking the limit as $i \to \infty$ we obtain the convergence of $(\|X_t - x^\star\|_t)_{t\in\mathbb{N}}$; more precisely, $\lim_{t\to\infty}\|X_t - x^\star\| = \lim_{i\to\infty} \nu_i$. We have thus proved $\mathcal{E}$ satisfies the requirements.

(3) *Conclude.* Combining the points (1) and (2), we get

$$
\mathbb{P}(\mathcal{E} \cap \{\liminf_{t\to\infty}\|V(X_t)\| = 0\}) = 1.
$$

Let us take a realization of this event. It holds $\liminf_{t\to\infty}\|V(X_t)\| = 0$ and we can thus extract a subsequence $(X_{\omega(t)})_{t\in\mathbb{N}}$ such that $\lim_{t\to\infty}\|V(X_{\omega(t)})\| = 0$. Let $x^\star \in \mathcal{X}^\star$, we know that $\|X_t - x^\star\|$ converges, implying that $(X_t)_{t\in\mathbb{N}}$ is bounded. As $\mathbb{R}^d$ is finite dimensional, we can then further extract $(X_{\omega(\psi(t))})_{t\in\mathbb{N}}$ so that $\lim_{t\to\infty} X_{\omega(\psi(t))} = x_\infty$ for some $x_\infty \in \mathbb{R}^d$. By continuity of $V$, we have $V(x_\infty) = 0$, i.e., $x_\infty \in \mathcal{X}^\star$. By the choice of $\mathcal{E}$, we have the convergence of $(\|X_t - x_\infty\|_t)_{t\in\mathbb{N}}$, and

$$
\lim_{t\to\infty}\|X_t - x_\infty\| = \lim_{t\to\infty}\|X_{\omega(\psi(t))} - x_\infty\| = \|x_\infty - x_\infty\| = 0.
$$

To conclude, we have proved that that $X_t$ converges to some $x^\star \in \mathcal{X}^\star$ almost surely. $\qquad\square$

### E.4 Proof of Theorem 3

**Theorem 3.** *Suppose that Assumptions 1–3 and 5 hold and assume that $\gamma_t \leq c/\beta$ with $c < 1$. Then:*

*1. If (DSEG) is run with $\gamma_t \equiv \gamma$, $\eta_t \equiv \eta$, we have:*

$$\mathbb{E}[\mathrm{dist}(X_t, \mathcal{X}^\star)^2] \leq (1 - \Delta)^{t-1} \, \mathrm{dist}(X_1, \mathcal{X}^\star)^2 + \frac{C}{\Delta}$$

*with constants $C = (2\gamma^2\eta\beta + \gamma^3\eta\beta^2 + \eta^2)\sigma^2$ and $\Delta = \gamma\eta\tau^2(1 - c^2)$.*

*2. If (DSEG) is run with $\gamma_t = \gamma/(t+b)^{1-\nu}$ and $\eta_t = \eta/(t+b)^\nu$ for some $\nu \in (1/2, 1)$, we have:*

$$\mathbb{E}[\mathrm{dist}(X_t, \mathcal{X}^\star)^2] \leq \frac{C}{\Delta - r}\frac{1}{t^r} + o\left(\frac{1}{t^r}\right)$$

*where $r = \min(1 - \nu, 2\nu - 1)$ and we further assume that $\gamma\eta\tau^2(1 - c^2) > r$. In particular, the optimal rate is attained when $\nu = 2/3$, which gives $\mathbb{E}[\mathrm{dist}(X_t, \mathcal{X}^\star)^2] = \mathcal{O}(1/t^{1/3})$.*

*For the sake of readability, the involved constants are stated for the case $\kappa = 0$. On the other hand, if $\sigma = 0$ and $\kappa \geq 0$, a geometric convergence can be proved.*

*Proof.* We first consider the case $\kappa = 0$ so that $\mathbb{E}[\|Z_t\|^2] \leq \sigma^2$ and $\mathbb{E}[\|Z_{t+\frac{1}{2}}\|^2] \leq \sigma^2$. Since $\gamma_t \leq c/\beta$, from (E.5) we deduce

$$\mathbb{E}_t[\|X_{t+1} - x^\star\|^2] \leq \|X_t - x^\star\|^2 - \gamma_t\eta_t(1 - c^2)\|V(X_t)\|^2 + (2\gamma_t^2\eta_t\beta + \gamma_t^3\eta_t\beta^2 + \eta_t^2)\sigma^2.$$

By concavity of the minimum operator, we then obtain

$$\begin{aligned}
\mathbb{E}_t\big[\min_{x^\star \in \mathcal{X}^\star} \|X_{t+1} - x^\star\|^2\big] &\leq \min_{x^\star \in \mathcal{X}^\star} \mathbb{E}_t[\|X_{t+1} - x^\star\|^2] \\
&\leq \min_{x^\star \in \mathcal{X}^\star} \|X_t - x^\star\|^2 - \gamma_t\eta_t(1 - c^2)\|V(X_t)\|^2 \\
&\quad + (2\gamma_t^2\eta_t\beta + \gamma_t^3\eta_t\beta^2 + \eta_t^2)\sigma^2.
\end{aligned}$$

In other words,

$$\mathbb{E}_t[\mathrm{dist}(X_{t+1}, \mathcal{X}^\star)^2] \leq \mathrm{dist}(X_t, \mathcal{X}^\star)^2 - \gamma_t\eta_t(1 - c^2)\|V(X_t)\|^2 + (2\gamma_t^2\eta_t\beta + \gamma_t^3\eta_t\beta^2 + \eta_t^2)\sigma^2.$$

Using Assumption 5 and the law of total expectation, this gives

$$\mathbb{E}[\mathrm{dist}(X_{t+1}, \mathcal{X}^\star)^2] \leq (1 - \gamma_t\eta_t\tau^2(1 - c^2))\, \mathbb{E}[\mathrm{dist}(X_t, \mathcal{X}^\star)^2] + (2\gamma_t^2\eta_t\beta + \gamma_t^3\eta_t\beta^2 + \eta_t^2)\sigma^2.$$

Points 1 and 2 are obtained respectively by applying Lemma D.1 and Lemma D.2.

For the case $\kappa \neq 0$, the term before $\mathbb{E}[\mathrm{dist}(X_t, \mathcal{X}^\star)^2]$ is replaced by $1 + C_t\kappa^2 - \rho_t\tau^2$ where $\rho_t = \gamma_t\eta_t - \gamma_t^3\eta_t\beta^2 - 8\gamma_t^2\eta_t^2\kappa^2$ is defined in the proof of Theorem 1. In point 1, the term $1 + C_t\kappa^2 - \rho_t\tau^2$ can be made in $(0, 1)$ for $\gamma$ and $\eta$ properly chosen. Precisely, we need

$$\left(4\gamma\beta + 2\gamma^2\beta^2 + \frac{4\eta}{\gamma} + 16\gamma\eta\,\kappa^2\right)\kappa^2 + (\gamma^2\beta^2 + 8\gamma\eta\,\kappa^2)\tau^2 < \tau^2.$$

To prove point 2, notice that the conditions of Lemma D.2 are still verified when $\gamma, \eta$ and $b$ are large enough. For example, if it holds for all $t$

$$\left(4\gamma_t\beta + 2\gamma_t^2\beta^2 + \frac{4\eta_t}{\gamma_t} + 16\gamma_t\eta_t\,\kappa^2\right)\kappa^2 + (\gamma_t^2\beta^2 + 8\gamma_t\eta_t\,\kappa^2)\tau^2 \leq \tau^2/2$$

and $\gamma\eta\tau^2/2 > r$ then Lemma D.2 can be applied. Finally, if $\sigma = 0$, the key inequality becomes

$$\mathbb{E}_t[\|X_{t+1} - x^\star\|^2] \leq (1 + C_t\kappa^2)\|X_t - x^\star\|^2 - \rho_t\|V(X_t)\|^2.$$

We therefore obtain geometric convergence for $1 + C_t\kappa^2 - \rho_t\tau^2 \in (0, 1)$. $\qquad\square$

## E.5 Proof of Theorem 5

**Theorem 5.** *Let $V$ be an affine operator satisfying Assumption 3, and suppose that Assumption 2 holds. Take a constant exploration stepsize $\gamma_t \equiv \gamma \le c/\beta$ with $c < 1$ (here $\beta$ is the largest singular value of the associated matrix). Then, the iterates $(X_t)_{t \in \mathbb{N}}$ of (DSEG) enjoy the following rates:*

*1. If the update stepsize is constant $\eta_t \equiv \eta \le \gamma$, then:*

$$\mathbb{E}[\operatorname{dist}(X_t, \mathcal{X}^\star)^2] \le (1 - \Delta)^{t-1} \operatorname{dist}(X_1, \mathcal{X}^\star)^2 + \frac{C}{\Delta}$$

*with $C = \eta^2(1 + c^2)\sigma^2$ and $\Delta = \gamma\eta\tau^2(1 - c^2)$.*

*2. If the update stepsize is of the form $\eta_t = \eta/(t + b)$ for $\eta > 1/(\tau^2\gamma(1 - c^2))$ and $b > \eta/\gamma$, then:*

$$\mathbb{E}[\operatorname{dist}(X_t, \mathcal{X}^\star)^2] \le \frac{C}{\Delta - 1}\frac{1}{t} + o\left(\frac{1}{t}\right).$$

For the sake of readability, the involved constants are stated for the case $\kappa = 0$.

*Proof.* To focus on the most important points of the proof, we shall consider the case $\kappa = 0$, while it is straightforward to derive the same kind of result when $\kappa > 0$ by following the reasoning of previous proofs. The crucial step here is then the derivation of a stochastic descent inequality in the form of (3). This is again based on (E.1). Writing $V(x) = Mx + v$, we can expand

$$\hat{V}_{t+\frac{1}{2}} = MX_t - \gamma_t M^2 X_t - \gamma_t Mv - \gamma_t MZ_t + v + Z_{t+\frac{1}{2}} = V(\tilde{X}_{t+\frac{1}{2}}) - \gamma_t MZ_t + Z_{t+\frac{1}{2}}.$$

We recall that $\tilde{X}_{t+\frac{1}{2}} = X_t - \gamma_t V(X_t)$. Let $x^\star \in \mathcal{X}^\star$. Together with the zero-mean assumption (1a), the above shows that

$$\mathbb{E}_t[\langle \hat{V}_{t+\frac{1}{2}}, \tilde{X}_{t+\frac{1}{2}} - x^\star \rangle] = \langle V(\tilde{X}_{t+\frac{1}{2}}), \tilde{X}_{t+\frac{1}{2}} - x^\star \rangle,$$

$$\mathbb{E}_t[\langle \hat{V}_{t+\frac{1}{2}}, V(X_t) \rangle] = \langle V(\tilde{X}_{t+\frac{1}{2}}), V(X_t) \rangle,$$

$$\mathbb{E}_t[\|\hat{V}_{t+\frac{1}{2}}\|^2] = \|V(\tilde{X}_{t+\frac{1}{2}})\|^2 + \mathbb{E}_t[\|\gamma_t MZ_t\|^2] + \mathbb{E}_t[\|Z_{t+\frac{1}{2}}\|^2].$$

Similar to (E.4), we write

$$-2\gamma_t\eta_t\langle V(\tilde{X}_{t+\frac{1}{2}}), V(X_t) \rangle + \eta_t^2\|V(\tilde{X}_{t+\frac{1}{2}})\|^2$$

$$\le -2\gamma_t\eta_t\langle V(\tilde{X}_{t+\frac{1}{2}}), V(X_t) \rangle + \gamma_t\eta_t\|V(\tilde{X}_{t+\frac{1}{2}})\|^2$$

$$= \gamma_t\eta_t(\|V(X_t) - V(\tilde{X}_{t+\frac{1}{2}})\|^2 - \|V(X_t)\|^2)$$

$$\le \gamma_t\eta_t(\gamma_t^2\beta^2 - 1)\|V(X_t)\|^2.$$

We have $\langle V(\tilde{X}_{t+\frac{1}{2}}), \tilde{X}_{t+\frac{1}{2}} - x^\star \rangle \ge 0$ by Assumption 3 and $\mathbb{E}_t[\|\gamma_t MZ_t\|^2] + \mathbb{E}_t[\|Z_{t+\frac{1}{2}}\|^2] \le (\gamma_t^2\beta^2 + 1)\sigma^2$ by Lipschitz continuity of $V$ and the finite variance assumption (i.e., (1b) with $\kappa = 0$). Taking expectation with respect to $\mathcal{F}_t$ over (E.1) then leads to

$$\mathbb{E}_t[\|X_{t+1} - x^\star\|^2] \le \|X_t - x^\star\|^2 - \gamma_t\eta_t(1 - \gamma_t^2\beta^2)\|V(X_t)\|^2 + \eta_t^2(\gamma_t^2\beta^2 + 1)\sigma^2$$

$$= \|X_t - x^\star\|^2 - \gamma_t\eta_t(1 - c^2)\|V(X_t)\|^2 + \eta_t^2(1 + c^2)\sigma^2.$$

Proceeding as in the proof of Theorem 3, we get

$$\mathbb{E}_t[\operatorname{dist}(X_{t+1}, \mathcal{X}^\star)^2] \le \operatorname{dist}(X_t, \mathcal{X}^\star)^2 - \gamma_t\eta_t(1 - c^2)\|V(X_t)\|^2 + \eta_t^2(1 + c^2)\sigma^2.$$

Since $V$ is affine, it verifies the error bound condition (EB). Writing $\gamma$ in the place of $\gamma_t$ and applying the law of total expectation, we obtain

$$\mathbb{E}[\operatorname{dist}(X_{t+1}, \mathcal{X}^\star)^2] \le (1 - \gamma\eta_t\tau^2(1 - c^2))\,\mathbb{E}[\operatorname{dist}(X_t, \mathcal{X}^\star)^2] + \eta_t^2(1 + c^2)\sigma^2.$$

We conclude with help of Lemma D.1 and Lemma D.2. $\qquad\square$

# F Proofs for local convergence results

## F.1 Local assumptions

For sake of clarity, we recall here the local assumptions that will bu used in the local convergence results.

**Assumption 1'.** The field $V$ is $\beta$-Lipschitz continuous near $x^\star$, i.e., for all $x, x'$ near $x^\star$,

$$\|V(x') - V(x)\| \leq \beta \|x' - x\|.$$

**Assumption 2'.** Let $x^\star \in \mathcal{X}^\star$ and $U$ be a neighborhood of $x^\star$. The noise term $Z_t$ of SFO satisfies

*a) Zero-mean:* $\qquad \mathbb{E}[Z_t \mid \mathcal{F}_t] \, \mathbb{1}_{\{X_t \in U\}} = 0.$ $\qquad\qquad\qquad\qquad$ (5a)

*b) Moment control:* $\quad \mathbb{E}[\|Z_t\|^q \mid \mathcal{F}_t] \, \mathbb{1}_{\{X_t \in U\}} \leq (\sigma + \kappa \|X_t - x^\star\|)^q.$ $\qquad$ (5b)

for some $q > 2$ and $\sigma, \kappa \geq 0$.

**Assumption 3'.** The operator $V$ satisfies $\langle V(x), x - x^\star \rangle \geq 0$ for all $x$ near $x^\star$.

**Assumption 5'.** $V$ is differentiable at $x^\star$ and its Jacobian matrix $\mathrm{Jac}_V(x^\star)$ is invertible.

For Assumption 2' in particular, when the neighborhood $U$ is bounded, the term $\kappa \|X_t - x^\star\|$ is also bounded and therefore, by choosing a larger $\sigma$ if needed, (5b) can be simplified to

$$\mathbb{E}[\|Z_t\|^q \mid \mathcal{F}_t] \, \mathbb{1}_{\{X_t \in U\}} \leq \sigma^q \quad \text{for all} \ \ x^\star \in \mathcal{X}^\star.$$

We will consider (5b) under this form in the sequel.

## F.2 Preparatory lemmas

The proofs of the local statements are much more demanding. The principle pillar of our analysis is a stability result formally stated in Appendix F.3. To prepare us for the challenge, we start by introducing the following lemma for bounding a recursive stochastic process.

**Lemma F.1.** *Consider a filtration $(\mathcal{F}_t)_{t \in \mathbb{N}}$ and four $(\mathcal{F}_t)_{t \in \mathbb{N}}$-adapted processes $(D_t)_{t \in \mathbb{N}}$, $(\zeta_t)_{t \in \mathbb{N}}$, $(\chi_t)_{t \in \mathbb{N}}$, $(\xi_t)_{t \in \mathbb{N}}$ such that $(\chi_t)_{t \in \mathbb{N}}$ is non-negative and the following recursive inequality is satisfied for all $t \geq 1$*

$$D_{t+1} \leq D_t - \zeta_t + \chi_{t+1} + \xi_{t+1}.$$

*Fixing a constant $C > 0$, we define the events $(A_t)_{t \in \mathbb{N}}$ by $A_1 := \{D_1 \leq C/2\}$ and $A_t := \{D_t \leq C\} \cap \{\chi_t \leq C/4\}$ for $t \geq 2$. We consider also the decreasing sequence of events $(I_t)_{t \in \mathbb{N}}$ defined by $I_t := \bigcap_{1 \leq s \leq t} A_s$. If the following three assumptions hold true*

*(i)* $\forall t, \zeta_t \, \mathbb{1}_{I_t} \geq 0,$

*(ii)* $\forall t, \mathbb{E}[\xi_{t+1} \mid \mathcal{F}_t] \, \mathbb{1}_{I_t} = 0,$

*(iii)* $\sum_{t=1}^{\infty} \mathbb{E}[(\xi_{t+1}^2 + \chi_{t+1}) \mathbb{1}_{I_t}] \leq \delta \varepsilon \, \mathbb{P}(A_1),$

*where $\varepsilon = \min(C^2/16, C/4)$ and $\delta \in (0, 1)$, then $\mathbb{P}\left(\bigcap_{t \geq 1} A_t \mid A_1\right) \geq 1 - \delta$.*

*Proof.* Let us start by introducing the following two $(\mathcal{F}_t)_{t \in \mathbb{N}}$-adapted submartingale sequences

$$S_t := \sum_{s=2}^{t} \xi_s \quad \text{and} \quad Q_t := S_t^2 + \sum_{s=2}^{t} \chi_s.$$

Subsequently, we define an auxiliary sequence of events

$$H_t := A_1 \cap \{ \max_{2 \leq s \leq t} Q_s \leq \varepsilon \}$$

which is also decreasing. With this at hand, we are ready to start our proof.

(1) *Inclusion $H_t \subset I_t$.* We prove the inclusion by induction. The statement is true when $t = 1$ as $H_1 = I_1 = A_1$. For $t \geq 2$, we write

$$D_t \leq D_1 - \sum_{s=1}^{t-1} \zeta_s + \sum_{s=2}^{t-1} \chi_{s+1} + \sum_{s=2}^{t-1} \xi_{s+1}. \qquad\qquad (\text{F.1})$$

By induction hypothesis, $H_{t-1} \subset I_{t-1}$, and thus for all $s \leq t-1$, we have $H_t \subset I_{t-1} \subset I_s$. Combining with *(i)* we deduce that for any realization of $H_t$, $\sum_{s=1}^{t-1} \zeta_s \geq 0$. On the other hand, by definition of $H_t$, it holds $Q_t \mathbb{1}_{H_t} \leq \varepsilon$. This implies

$$\left(\sum_{s=2}^{t-1} \xi_{s+1}\right) \mathbb{1}_{H_t} = S_t \mathbb{1}_{H_t} \leq \sqrt{\varepsilon} \leq C/4, \tag{F.2}$$

$$\left(\sum_{s=2}^{t-1} \chi_{s+1}\right) \mathbb{1}_{H_t} \leq \varepsilon \leq C/4. \tag{F.3}$$

Finally as $H_t \subset A_1$ we have $D_1 \mathbb{1}_{H_t} \leq C/2$. Therefore, for any realization of $H_t$, using (F.1) gives

$$D_t \leq C/2 - 0 + C/4 + C/4 = C.$$

In the meantime (F.2) ensures as well $\chi_t \mathbb{1}_{H_t} \leq C/4$ and we have thus proven $H_t \subset A_t$. Using $H_t \subset H_{t-1} \subset I_{t-1}$, we conclude $H_t \subset I_t$.

*(2) Recursive bound on* $\mathbb{E}[Q_t \mathbb{1}_{H_{t-1}}]$. Since $H_{t-1} \subseteq H_{t-2}$, it holds $H_{t-1} = H_{t-2} \setminus (H_{t-2} \setminus H_{t-1})$. We can therefore decompose

$$\begin{aligned}\mathbb{E}[Q_t \mathbb{1}_{H_{t-1}}] &= \mathbb{E}[(Q_t - Q_{t-1}) \mathbb{1}_{H_{t-1}}] + \mathbb{E}[Q_{t-1} \mathbb{1}_{H_{t-1}}] \\ &= \mathbb{E}[(\xi_t^2 + 2\xi_t S_{t-1} + \chi_t) \mathbb{1}_{H_{t-1}}] + \mathbb{E}[Q_{t-1} \mathbb{1}_{H_{t-2}}] - \mathbb{E}[Q_{t-1} \mathbb{1}_{H_{t-2} \setminus H_{t-1}}].\end{aligned}$$

From the law of total expectation, $H_{t-1} \subset I_{t-1}$ and *(ii)* we have

$$\mathbb{E}[\xi_t S_{t-1} \mathbb{1}_{H_{t-1}}] = \mathbb{E}[\mathbb{E}[\xi_t \mid \mathcal{F}_{t-1}] S_{t-1} \mathbb{1}_{H_{t-1}}] = 0.$$

As $\xi_t^2 + \chi_t$ is non-negative, using again $H_{t-1} \subset I_{t-1}$, we get

$$\mathbb{E}[(\xi_t^2 + \chi_t) \mathbb{1}_{H_{t-1}}] \leq \mathbb{E}[(\xi_t^2 + \chi_t) \mathbb{1}_{I_{t-1}}].$$

By definition for any realization in $H_{t-2} \setminus H_{t-1}$, it holds $Q_{t-1} > \varepsilon$ and thus

$$\mathbb{E}[Q_{t-1} \mathbb{1}_{H_{t-2} \setminus H_{t-1}}] \geq \varepsilon \mathbb{E}[\mathbb{1}_{H_{t-2} \setminus H_{t-1}}] = \varepsilon \mathbb{P}(H_{t-2} \setminus H_{t-1}).$$

Combining the above we deduce the following recursive bound

$$\mathbb{E}[Q_t \mathbb{1}_{H_{t-1}}] \leq \mathbb{E}[Q_{t-1} \mathbb{1}_{H_{t-2}}] + \mathbb{E}[(\xi_t^2 + \chi_t) \mathbb{1}_{I_{t-1}}] - \varepsilon \mathbb{P}(H_{t-2} \setminus H_{t-1}). \tag{F.4}$$

*(3) Conclude.* Summing (F.4) from $t = 3$ to $T$ we obtain

$$\begin{aligned}\mathbb{E}[Q_T \mathbb{1}_{H_{T-1}}] &\leq \mathbb{E}[Q_2 \mathbb{1}_{H_1}] + \sum_{t=3}^{T} \mathbb{E}[(\xi_t^2 + \chi_t) \mathbb{1}_{I_{t-1}}] - \varepsilon \sum_{t=3}^{T} \mathbb{P}(H_{t-2} \setminus H_{t-1}) \\ &= \sum_{t=2}^{T} \mathbb{E}[(\xi_t^2 + \chi_t) \mathbb{1}_{I_{t-1}}] - \varepsilon \mathbb{P}(A_1 \setminus H_{T-1}),\end{aligned} \tag{F.5}$$

where in the second line we use $Q_2 = \xi_2^2 + \chi_2$, $H_1 = I_1 = A_1$ and $H_1 \setminus H_{T-1} = \dot{\bigcup}_{3 \leq t \leq T}(H_{t-2} \setminus H_{t-1})$ with $\dot{\bigcup}$ denoting the disjoint union (true since $(H_t)_{t \geq 1}$ is a decreasing sequence of events). By repeating the same arguments that are used before and using the fact that $Q_T$ is non-negative,

$$\begin{aligned}\mathbb{P}(A_1 \setminus H_T) &= \mathbb{P}(H_{T-1} \setminus H_T) + \mathbb{P}(A_1 \setminus H_{T-1}) \\ &\leq \frac{1}{\varepsilon} \mathbb{E}[Q_T \mathbb{1}_{H_{T-1} \setminus H_T}] + \mathbb{P}(A_1 \setminus H_{T-1}) \\ &\leq \frac{1}{\varepsilon} \mathbb{E}[Q_T \mathbb{1}_{H_{T-1}}] + \mathbb{P}(A_1 \setminus H_{T-1}).\end{aligned} \tag{F.6}$$

(F.6), (F.5) along with *(iii)* lead to

$$\mathbb{P}(A_1 \setminus H_T) \leq \frac{1}{\varepsilon} \sum_{t=2}^{T} \mathbb{E}[(\xi_t^2 + \chi_t) \mathbb{1}_{I_{t-1}}] \leq \delta \mathbb{P}(A_1).$$

Subsequently,

$$\mathbb{P}(H_T \mid A_1) = 1 - \frac{\mathbb{P}(A_1 \setminus H_T)}{\mathbb{P}(A_1)} \geq 1 - \delta.$$

With $H_T \subset I_T$ this also gives $\mathbb{P}(I_T \mid A_1) \geq 1 - \delta$. We notice that $\bigcap_{t \geq 1} I_t = \bigcap_{t \geq 1} A_t$. As $(I_t)_{t \geq 1}$ is decreasing, by continuity from above we conclude

$$\mathbb{P}\left(\bigcap_{t \geq 1} A_t \mid A_1\right) = \lim_{t \to \infty} \mathbb{P}(I_t \mid A_1) \geq 1 - \delta.$$

$\square$

To apply Lemma F.1, we establish another quasi-descent lemma which holds without taking expectation values.

**Lemma F.2.** *For all $x^\star \in \mathcal{X}^\star$, $t \in \mathbb{N}$, the iterates generated by* (DSEG) *satisfy the following inequality*

$$\|X_{t+1} - x^\star\|^2 \leq \|X_t - x^\star\|^2 - 2\eta_t \langle V(X_{t+\frac{1}{2}}), X_{t+\frac{1}{2}} - x^\star \rangle$$
$$- 2\gamma_t \eta_t \|V(X_t)\|(\|V(X_t)\| - \|V(\tilde{X}_{t+\frac{1}{2}}) - V(X_t)\|)$$
$$- 2\eta_t \langle Z_{t+\frac{1}{2}}, X_t - x^\star \rangle - 2\gamma_t \eta_t \langle V(\tilde{X}_{t+\frac{1}{2}}), Z_t \rangle$$
$$+ 2\gamma_t \eta_t \|\hat{V}_t\| \|V(X_{t+\frac{1}{2}}) - V(\tilde{X}_{t+\frac{1}{2}})\| + \eta_t^2 \|\hat{V}_{t+\frac{1}{2}}\|^2 \qquad \text{(F.7)}$$

*If we assume* Assumption 1' *for some solution $x^\star$ and that $X_t$, $\tilde{X}_{t+\frac{1}{2}}$, $X_{t+\frac{1}{2}}$ all lie in this neighborhood, then*

$$\|X_{t+1} - x^\star\|^2 \leq \|X_t - x^\star\|^2 - 2\eta_t \langle V(X_{t+\frac{1}{2}}), X_{t+\frac{1}{2}} - x^\star \rangle - 2\gamma_t \eta_t (1 - \gamma_t \beta) \|V(X_t)\|^2$$
$$- 2\eta_t \langle Z_{t+\frac{1}{2}}, X_t - x^\star \rangle - 2\gamma_t \eta_t \langle V(\tilde{X}_{t+\frac{1}{2}}), Z_t \rangle + 2\gamma_t^2 \eta_t \beta \|Z_t\| \|\hat{V}_t\| + \eta_t^2 \|\hat{V}_{t+\frac{1}{2}}\|^2. \qquad \text{(F.8)}$$

*Proof.* Similar to (E.1), we develop

$$\|X_{t+1} - x^\star\|^2 = \|X_t - x^\star\|^2 - 2\eta_t \langle V(X_{t+\frac{1}{2}}), X_t - x^\star \rangle - 2\eta_t \langle Z_{t+\frac{1}{2}}, X_t - x^\star \rangle + \eta_t^2 \|\hat{V}_{t+\frac{1}{2}}\|^2.$$

We further develop the second term on the RHS of the equality

$$\langle V(X_{t+\frac{1}{2}}), X_t - x^\star \rangle = \langle V(X_{t+\frac{1}{2}}), X_{t+\frac{1}{2}} - x^\star \rangle + \gamma_t \langle V(X_{t+\frac{1}{2}}), \hat{V}_t \rangle$$
$$= \langle V(X_{t+\frac{1}{2}}), X_{t+\frac{1}{2}} - x^\star \rangle + \gamma_t \langle V(X_{t+\frac{1}{2}}) - V(\tilde{X}_{t+\frac{1}{2}}), \hat{V}_t \rangle + \gamma_t \langle V(\tilde{X}_{t+\frac{1}{2}}), \hat{V}_t \rangle.$$

To deal with the last term

$$\langle V(\tilde{X}_{t+\frac{1}{2}}), \hat{V}_t \rangle = \langle V(\tilde{X}_{t+\frac{1}{2}}), V(X_t) \rangle + \langle V(\tilde{X}_{t+\frac{1}{2}}), Z_t \rangle$$
$$= \langle V(\tilde{X}_{t+\frac{1}{2}}) - V(X_t), V(X_t) \rangle + \|V(X_t)\|^2 + \langle V(\tilde{X}_{t+\frac{1}{2}}), Z_t \rangle.$$

By combing all the above, we readily get (F.7) with Cauchy's inequality. If Assumption 1' holds on a set that $X_t$, $\tilde{X}_{t+\frac{1}{2}}$, $X_{t+\frac{1}{2}}$ belong to, we can further bound

$$2\gamma_t \eta_t \|V(X_{t+\frac{1}{2}}) - V(\tilde{X}_{t+\frac{1}{2}})\| \|\hat{V}_t\| \leq 2\gamma_t^2 \eta_t \beta \|Z_t\| \|\hat{V}_t\|,$$
$$2\gamma_t \eta_t \|V(\tilde{X}_{t+\frac{1}{2}}) - V(X_t)\| \|V(X_t)\| \leq 2\gamma_t^2 \eta_t \beta \|V(X_t)\|^2,$$

which gives (F.8). $\square$

### F.3  A stability result

The following theorem characterizes the stability of the algorithm around a solution. The subsequent stepsize condition encompasses the stepsizes employed in Theorem 2 and Theorem 4 as special cases. We recall that $\tilde{X}_{t+\frac{1}{2}} = X_t - \gamma_t V(X_t)$.

**Theorem F.1.** *Let $x^\star$ be an isolated solution of* (Opt) *such that Assumptions $1'$–$3'$ are satisfied on* $\mathbb{B}_r(x^\star)$ *for some $q > 2, r > 0$. We fix a tolerance level $\delta \in (0,1)$. For every $\rho \in (0,1)$, there is a neighborhood $U_\rho$ of $x^\star$ and a constant $\Gamma > 0$ such that if* (DSEG) *is initialized at $X_1 \in U_\rho$ and is run with stepsizes satisfying $\sum_t \gamma_t \eta_t = \infty$, $\sum_t \eta_t^2 < \Gamma$, $\sum_t \gamma_t^2 \eta_t < \Gamma$ and $\sum_t \gamma_t^q < \Gamma$, then*

$$E_\infty^\rho = \{X_{t+\frac{1}{2}} \in \mathbb{B}_r(x^\star), X_t, \tilde{X}_{t+\frac{1}{2}} \in \mathbb{B}_{\rho r}(x^\star) \text{ for all } t = 1, 2, \dots\}$$

*occurs with probability at least $1 - \delta$, i.e., $\mathbb{P}(E_\infty^\rho \mid X_1 \in U_\rho) \geq 1 - \delta$.*

*Proof.* We would like to apply Lemma F.1, but instead of indexing by $t \in \mathbb{N}$, we index by $s \in \mathbb{N}/2$. We invoke (F.7) from Lemma F.2 and set the random variables accordingly

$$\underbrace{\|X_{t+1} - x^\star\|^2}_{D_{t+1}} \leq \underbrace{\|X_t - x^\star\|^2}_{D_t} \underbrace{- 2\eta_t \langle V(X_{t+\frac{1}{2}}), X_{t+\frac{1}{2}} - x^\star\rangle}_{\zeta_{t+\frac{1}{2}}}$$

$$\underbrace{- 2\gamma_t \eta_t \|V(X_t)\|(\|V(X_t)\| - \|V(\tilde{X}_{t+\frac{1}{2}}) - V(X_t)\|)}_{\zeta_t}$$

$$+ \underbrace{(-2\eta_t \langle Z_{t+\frac{1}{2}}, X_t - x^\star\rangle)}_{\xi_{t+1}} + \underbrace{(-2\gamma_t \eta_t \langle V(\tilde{X}_{t+\frac{1}{2}}), Z_t\rangle)}_{\xi_{t+\frac{1}{2}}}$$

$$+ \underbrace{2\gamma_t \eta_t \|\hat{V}_t\|\|V(X_{t+\frac{1}{2}}) - V(\tilde{X}_{t+\frac{1}{2}})\| + \eta_t^2 \|\hat{V}_{t+\frac{1}{2}}\|^2}_{\chi_{t+1}} \qquad (\text{F.9})$$

We additionally define $\chi_{t+\frac{1}{2}} := \gamma_t^q \|Z_t\|^q$ and $D_{t+\frac{1}{2}} := D_t - \zeta_t + \chi_{t+\frac{1}{2}} + \xi_{t+\frac{1}{2}}$ so that (F.9) implies $D_{t+1} \leq D_{t+\frac{1}{2}} - \zeta_{t+\frac{1}{2}} + \chi_{t+1} + \xi_{t+1}$. With the definition of $D_{t+\frac{1}{2}}$ the inequality (F.1) is indeed checked with all half integers. We should now verify that the assumptions *(i)*, *(ii)* and *(iii)* in Lemma F.1 are satisfied for a $C$ that is properly chosen. Let $M$ denote the supremum of $\|V(x)\|$ for $x \in U'$ where $U' = \mathbb{B}_{r'}(x^\star)$ and $r' := \rho r$. We then choose $C := \min(r'^2/9, 4(r'/3)^q)$. We also set $\Gamma$ small enough to guarantee $\gamma_t \leq \min(r'/(3M), 1/\beta)$.

(a.0) *Inclusion $I_t \subset \{X_t, \tilde{X}_{t+\frac{1}{2}} \in U'\}$ and $I_{t+\frac{1}{2}} \subset \{X_t, \tilde{X}_{t+\frac{1}{2}}, X_{t+\frac{1}{2}} \in U'\}$.* Since $I_t \subset A_t$, for any realization of $I_t$, we have $\|X_t - x^\star\|^2 \leq C \leq r'^2/9$. It follows

$$\|\tilde{X}_{t+\frac{1}{2}} - x^\star\|^2 \leq 2\|X_t - x^\star\|^2 + 2\gamma_t^2 \|V(X_t)\|^2 \leq \frac{2r'^2}{9} + 2\gamma_t^2 M^2 \leq \frac{4r'^2}{9}.$$

We have shown $I_t \subset \{X_t, \tilde{X}_{t+\frac{1}{2}} \in U'\}$. On the other hand, $I_{t+\frac{1}{2}} \subset A_t \cap A_{t+\frac{1}{2}} \subset \{D_t \leq C\} \cap \{\chi_{t+\frac{1}{2}} \leq C/4\}$. Therefore for any realization of $I_{t+\frac{1}{2}}$,

$$\gamma_t^q \|Z_t\|^q = \chi_{t+\frac{1}{2}} \leq \frac{C}{4} \leq (r'/3)^q.$$

Subsequently,

$$\|X_{t+\frac{1}{2}} - x^\star\|^2 \leq 3\|X_t - x^\star\|^2 + 3\gamma_t^2 \|V(X_t)\|^2 + 3\gamma_t^2 \|Z_t\|^2 \leq \frac{r'^2}{3} + \frac{r'^2}{3} + 3\left(\frac{r'}{3}\right)^2 \leq r'^2.$$

This proves $I_{t+\frac{1}{2}} \subset \{X_t, \tilde{X}_{t+\frac{1}{2}}, X_{t+\frac{1}{2}} \in U'\}$.

(a.1) *Assumption (i).* We start by $\zeta_{t+\frac{1}{2}} \mathbb{1}_{I_{t+\frac{1}{2}}} \geq 0$. This is true because $I_{t+\frac{1}{2}} \subset \{X_{t+\frac{1}{2}} \in U'\}$ and $U' \subset \mathbb{B}_r(x^\star)$, which allows us to apply Assumption $3'$ to obtain $\langle V(X_{t+\frac{1}{2}}), X_{t+\frac{1}{2}} - x^\star\rangle \geq 0$ whenever $I_{t+\frac{1}{2}}$ occurs. Similarly, by $I_t \subset \{X_t, \tilde{X}_{t+\frac{1}{2}} \in U'\}$ and Assumption $1'$ we then have

$$\zeta_t \mathbb{1}_{I_t} \geq 2\gamma_t \eta_t (1 - \gamma_t \beta)\|V(X_t)\|^2 \geq 0.$$

(a.2) *Assumption (ii).* Immediate from (5a), (a.0) and the law of the total expectation.

(a.3) *Assumption (iii).* By using that $I_t \subset \{\tilde{X}_{t+\frac{1}{2}} \in U'\}$ and $I_t \subset \{X_t \in \mathbb{B}_r(x^\star)\}$, we get

$$\mathbb{E}[\xi_{t+\frac{1}{2}}^2 \, \mathbb{1}_{I_t}] \leq 4\gamma_t^2 \eta_t^2 \, \mathbb{E}[\|V(\tilde{X}_{t+\frac{1}{2}})\|^2 \, \mathbb{1}_{I_t} \|Z_t\|^2 \, \mathbb{1}_{I_t}]$$
$$\leq 4\gamma_t^2 \eta_t^2 M^2 \, \mathbb{E}[\|Z_t\|^2 \, \mathbb{1}_{\{X_t \in \mathbb{B}_r(x^\star)\}}] \leq 4\gamma_t^2 \eta_t^2 M^2 \sigma^2.$$

For the last inequality we use (5b) and Jensen's inequality to bound $\mathbb{E}[\|Z_t\|^2 \, \mathbb{1}_{\{X_t \in \mathbb{B}_r(x^\star)\}}]$. Similarly,

$$\mathbb{E}[\|Z_t\| \, \mathbb{1}_{\{X_t \in \mathbb{B}_r(x^\star)\}}] \leq \sigma,$$
$$\mathbb{E}[\|Z_{t+\frac{1}{2}}\|^2 \, \mathbb{1}_{\{X_{t+\frac{1}{2}} \in \mathbb{B}_r(x^\star)\}}] \leq \sigma^2.$$

Using $I_{t+\frac{1}{2}} \subset \{X_t, \tilde{X}_{t+\frac{1}{2}}, X_{t+\frac{1}{2}} \in U'\}$ and Assumption 1' then gives

$$\mathbb{E}[\chi_{t+1} \, \mathbb{1}_{I_{t+\frac{1}{2}}}] \leq 2\gamma_t^2 \eta_t \beta \, \mathbb{E}[\|Z_t\|(\|V(X_t)\| + \|Z_t\|) \, \mathbb{1}_{I_{t+\frac{1}{2}}}]$$
$$+ \eta_t^2 \, \mathbb{E}[(\|V(X_{t+\frac{1}{2}})\|^2 + \|Z_{t+\frac{1}{2}}\|^2) \, \mathbb{1}_{I_{t+\frac{1}{2}}}]$$
$$\leq 2\gamma_t^2 \eta_t \beta (\mathbb{E}[\|Z_t\|^2 \, \mathbb{1}_{\{X_t \in \mathbb{B}_r(x^\star)\}}] + \mathbb{E}[\|Z_t\| \, \mathbb{1}_{\{X_t \in \mathbb{B}_r(x^\star)\}} \|V(X_t)\| \, \mathbb{1}_{\{X_t \in U'\}}])$$
$$+ \eta_t^2 (\mathbb{E}[\|V(X_{t+\frac{1}{2}})\|^2 \, \mathbb{1}_{\{X_{t+\frac{1}{2}} \in U'\}}] + \mathbb{E}[\|Z_{t+\frac{1}{2}}\|^2 \, \mathbb{1}_{\{X_{t+\frac{1}{2}} \in \mathbb{B}_r(x^\star)\}}])$$
$$\leq 2\gamma_t^2 \eta_t \beta (M\sigma + \sigma^2) + \eta_t^2 (M^2 + \sigma^2). \tag{F.10}$$

By similar arguments and in particular by invoking $I_{t+\frac{1}{2}} \subset \{D_t \leq C\}$ and the definition of $C$, it follows

$$\mathbb{E}[\xi_{t+1}^2 \, \mathbb{1}_{I_{t+\frac{1}{2}}}] \leq \frac{4}{9} \eta_t^2 r'^2 \sigma^2,$$

Combining the above with $\mathbb{E}[\chi_{t+\frac{1}{2}} \, \mathbb{1}_{I_t}] \leq \gamma_t^q \sigma^q$, we have

$$\sum_{s \in 1, 3/2, \dots} (\xi_{s+\frac{1}{2}}^2 + \chi_{s+\frac{1}{2}}) \, \mathbb{1}_{I_s}$$

$$\leq \sum_{t=1}^{\infty} \left( \gamma_t^q \sigma^q + 2\gamma_t^2 \eta_t \beta (M\sigma + \sigma^2) + 4\gamma_t^2 \eta_t^2 M^2 \sigma^2 + \eta_t^2 (M^2 + \sigma^2 + \frac{4}{9} r'^2 \sigma^2) \right)$$

$$\leq \left( \sigma^q + 2\beta(M\sigma + \sigma^2) + \frac{4}{\beta} M^2 \sigma^2 + M^2 + \sigma^2 + \frac{4}{9} r'^2 \sigma^2 \right) \Gamma.$$

We can thus pick $\Gamma$ small enough to make *(iii)* verified.

(a.4) *Conclude.* We set $U_\rho = \mathbb{B}_{\sqrt{C/2}}(x^\star)$ so that $A_1 = \{X_1 \in U_\rho\}$. By invoking Lemma F.1 we get $\mathbb{P}\left( \bigcap_{t \geq 1} A_t \mid A_1 \right) \geq 1 - \delta$. Additionally, (a.0) along with $U' \subset \mathbb{B}_r(x^\star)$ imply $\bigcap_{t \geq 1} A_t \subset E_\infty^\rho$, concluding the proof. $\qquad\qquad\square$

## F.4 Proof of Theorem 2

**Theorem 2.** *Fix a tolerance level $\delta > 0$ and suppose that Assumptions 1'–3' hold for some isolated solution $x^\star$ of (Opt). Assume further that (DSEG) is run with stepsize parameters of the form (4) with small enough $\gamma$, $\eta$ and proper choice of $r_\gamma, r_\eta$ (cf. Fig. 2). If the algorithm is not initialized too far from $x^\star$, its iterates converge to $x^\star$ with probability at least $1 - \delta$.*

*Proof.* Let $r > 0$, $\rho \in (0, 1)$. By Theorem F.1, we know that if (DSEG) is run as stated in Theorem 2 with $r_\gamma + r_\eta \leq 1$, $2r_\eta > 1$, $2r_\gamma + r_\eta > 1$, $r_\gamma q > 1$ and small enough $\gamma, \eta$, the event $E_\infty^\rho$ occurs with probability $1 - \delta$. With this at hand we are ready to prove the large probability convergence result. For $t \in \mathbb{N}$, let us define the following events

$$E_t := \{X_s, \tilde{X}_{s+\frac{1}{2}} \in \mathbb{B}_{\rho r}(x^\star) \text{ for all } s = 1, 2, \dots, t\}$$
$$E_{t+\frac{1}{2}} := E_t \cap \{X_{s+\frac{1}{2}} \in \mathbb{B}_r(x^\star) \text{ for all } s = 1, 2, \dots, t\}.$$

We notice that $E_\infty^\rho = \bigcap_{t \geq 1} E_{t+\frac{1}{2}}$. We would like to establish a recursive inequality in the form of (D.1) by taking $U_t = \|X_t - x^\star\| \mathbb{1}_{E_{t-\frac{1}{2}}}$. The main difficulty consists in controlling the term

$\mathbb{E}_t[\langle V(\tilde{X}_{t+\frac{1}{2}}), Z_t\rangle \, \mathbb{1}_{E_{t+\frac{1}{2}}}]$, which is generally non-zero as $\mathbb{1}_{E_{t+\frac{1}{2}}}$ is not $\mathcal{F}_t$-measurable. To achieve this, we rely on the following key observation.

$$\mathbb{E}_t[Z_t \, \mathbb{1}_{E_t}] = \mathbb{E}_t[Z_t \, \mathbb{1}_{E_{t+\frac{1}{2}}}] + \mathbb{E}_t[Z_t \, \mathbb{1}_{E_t \setminus E_{t+\frac{1}{2}}}].$$

As $\mathbb{1}_{E_t}$ is $\mathcal{F}_t$-measurable and $E_t \subset \{X_t \in \mathbb{B}_r(x^\star)\}$, $\mathbb{E}_t[Z_t \, \mathbb{1}_{E_t}]$ is indeed zero and this implies

$$\|\mathbb{E}_t[Z_t \, \mathbb{1}_{E_{t+\frac{1}{2}}}]\| = \|\mathbb{E}_t[Z_t \, \mathbb{1}_{E_t \setminus E_{t+\frac{1}{2}}}]\|. \tag{F.11}$$

The problem then reduces to finding an upper bound of $\|\mathbb{E}_t[Z_t \, \mathbb{1}_{E_t \setminus E_{t+\frac{1}{2}}}]\|$. By definition, for any realization of $E_t \setminus E_{t+\frac{1}{2}}$, $\tilde{X}_{t+\frac{1}{2}} \in \mathbb{B}_{\rho r}(x^\star)$ and $X_{t+\frac{1}{2}} \notin \mathbb{B}_r(x^\star)$. Since $X_{t+\frac{1}{2}} = \tilde{X}_{t+\frac{1}{2}} - \gamma_t Z_t$, we deduce

$$E_t \setminus E_{t+\frac{1}{2}} \subset \{\|\gamma_t Z_t\| \geq (1-\rho)r\}.$$

Therefore, using $E_t \subset \{X_t \in \mathbb{B}_r(x^\star)\}$ along with the Chebyshev's inequality yields

$$\mathbb{P}(E_t \setminus E_{t+\frac{1}{2}} \mid \mathcal{F}_t) \leq \mathbb{P}\left(\|Z_t\| \mathbb{1}_{\{X_t \in \mathbb{B}_r(x^\star)\}} \geq \frac{(1-\rho)r}{\gamma_t} \mid \mathcal{F}_t\right) \leq \frac{\sigma^2 \gamma_t^2}{(1-\rho)^2 r^2}.$$

Applying the Cauchy–Schwarz inequality leads to

$$\|\mathbb{E}_t[Z_t \, \mathbb{1}_{E_t \setminus E_{t+\frac{1}{2}}}]\| \leq \sqrt{\mathbb{E}_t[\|Z_t \, \mathbb{1}_{E_t}\|^2]} \sqrt{\mathbb{E}_t[\mathbb{1}_{E_t \setminus E_{t+\frac{1}{2}}}^2]} \leq \frac{\sigma^2 \gamma_t}{(1-\rho)r}. \tag{F.12}$$

Then, by using (F.11), (F.12) and $E_{t+\frac{1}{2}} \subset E_t$,

$$\begin{aligned}
\mathbb{E}_t[\langle V(\tilde{X}_{t+\frac{1}{2}}), Z_t\rangle \, \mathbb{1}_{E_{t+\frac{1}{2}}}] &= \mathbb{E}_t[\langle V(\tilde{X}_{t+\frac{1}{2}}) \, \mathbb{1}_{E_t}, Z_t \, \mathbb{1}_{E_{t+\frac{1}{2}}}\rangle] \\
&= \langle V(\tilde{X}_{t+\frac{1}{2}}) \, \mathbb{1}_{E_t}, \mathbb{E}_t[Z_t \, \mathbb{1}_{E_{t+\frac{1}{2}}}]\rangle \\
&\leq \|V(\tilde{X}_{t+\frac{1}{2}}) \, \mathbb{1}_{E_t}\| \|\mathbb{E}_t[Z_t \, \mathbb{1}_{E_{t+\frac{1}{2}}}]\| \\
&\leq \frac{M\sigma^2 \gamma_t}{(1-\rho)r}, 
\end{aligned} \tag{F.13}$$

where $M := \sup_{x \in \mathbb{B}_r(x^\star)} \|V(x)\|$. We can now derive a recursive bound on $\mathbb{E}[\|X_{t+1} - x^\star\| \, \mathbb{1}_{E_{t+\frac{1}{2}}}]$ by invoking Lemma F.2. The inequality (F.8) multiplied by $\mathbb{1}_{E_{t+\frac{1}{2}}}$ holds true by definition of $E_{t+\frac{1}{2}}$ and Assumption 1'. The desired inequality can then be obtained by taking expectation conditioned on $\mathcal{F}_t$. On the one hand, we use

$$\langle V(X_{t+\frac{1}{2}}), X_{t+\frac{1}{2}} - x^\star\rangle \, \mathbb{1}_{E_{t+\frac{1}{2}}} \geq 0$$

$$\mathbb{E}_t[\langle Z_{t+\frac{1}{2}}, X_t - x^\star\rangle \, \mathbb{1}_{E_{t+\frac{1}{2}}}] = \mathbb{E}_t[\langle \mathbb{E}_{t+\frac{1}{2}}[Z_{t+\frac{1}{2}}] \, \mathbb{1}_{E_{t+\frac{1}{2}}}, X_t - x^\star\rangle] = 0.$$

On the other hand, the last two terms of (F.8) can be bounded similarly as in (F.10) and the antepenultimate term can now be bounded thanks to (F.13). We then obtain

$$\begin{aligned}
\mathbb{E}_t[\|X_{t+1} - x^\star\|^2 \, \mathbb{1}_{E_{t+\frac{1}{2}}}] &\leq \mathbb{E}_t[\|X_t - x^\star\|^2 \, \mathbb{1}_{E_{t+\frac{1}{2}}}] - 0 - 2\gamma_t \eta_t (1 - \gamma_t \beta) \, \mathbb{E}_t[\|V(X_t)\|^2 \, \mathbb{1}_{E_{t+\frac{1}{2}}}] \\
&\quad - 0 + 2\gamma_t^2 \eta_t \frac{M\sigma^2}{(1-\rho)r} + \eta_t^2 (M^2 + \sigma^2) + 2\gamma_t^2 \eta_t \beta (M\sigma + \sigma^2). 
\end{aligned} \tag{F.14}$$

Without loss of generality we may suppose $\gamma_t \beta \leq 1/2$. To simplify the notation, we set

$$\zeta_t = \min\left(\|X_t - x^\star\|^2, \gamma_t \eta_t \|V(X_t)\|^2\right), \quad \mathcal{M}_1 = 2\frac{M\sigma^2}{(1-\rho)r} + 2\beta(M\sigma + \sigma^2), \quad \mathcal{M}_2 = M^2 + \sigma^2.$$

It follows from (F.14)

$$\mathbb{E}_t[\|X_{t+1} - x^\star\|^2 \, \mathbb{1}_{E_{t+\frac{1}{2}}}] \leq \mathbb{E}_t[(\|X_t - x^\star\|^2 - \zeta_t) \, \mathbb{1}_{E_{t+\frac{1}{2}}}] + \gamma_t^2 \eta_t \mathcal{M}_1 + \eta_t^2 \mathcal{M}_2.$$

As $\|X_t - x^\star\|^2 - \zeta_t \geq 0$ and $E_{t+\frac{1}{2}} \subset E_{t-\frac{1}{2}}$, this implies

$$\mathbb{E}_t[\|X_{t+1} - x^\star\|^2 \, \mathbb{1}_{E_{t+\frac{1}{2}}}] \leq \|X_t - x^\star\|^2 \, \mathbb{1}_{E_{t-\frac{1}{2}}} - \zeta_t \, \mathbb{1}_{E_{t-\frac{1}{2}}} + \gamma_t^2 \eta_t \mathcal{M}_1 + \eta_t^2 \mathcal{M}_2.$$

Invoking the Robbins–Siegmund theorem (Lemma D.4) gives the almost sure convergence of $\sum_t \zeta_t \mathbb{1}_{E_{t-\frac{1}{2}}}$ and $\|X_t - x^\star\|^2 \mathbb{1}_{E_{t-\frac{1}{2}}}$. We use $\mathbb{P}(E_\infty^\rho) > 1 - \delta$ and deduce that

$$
\mathbb{P}\left( \underbrace{E_\infty^\rho \cap \left\{ \sum_{t=1}^\infty \zeta_t \mathbb{1}_{E_{t-\frac{1}{2}}} < \infty \right\} \cap \left\{ \|X_t - x^\star\|^2 \mathbb{1}_{E_{t-\frac{1}{2}}} \text{ converges} \right\}}_{\mathcal{E}} \right) \geq 1 - \delta.
$$

Since $E_\infty^\rho = \bigcap_{t \geq 1} E_{t+\frac{1}{2}}$, for any realization of the above event it holds $\sum_t \zeta_t < \infty$ and $\|X_t - x^\star\|^2$ converges. We assume by contradiction that $\|X_t - x^\star\|^2$ converges to some constant $\nu > 0$. From the summability of $(\zeta_t)_{t \in \mathbb{N}}$ we know that $\zeta_t \to 0$ and therefore for all $t$ large enough we have in fact $\zeta_t = \gamma_t \eta_t \|V(X_t)\|^2$. It follows that $\sum_t \gamma_t \eta_t \|V(X_t)\|^2 < \infty$. Repeating the arguments of Appendix E.3 (Proof of Theorem 1) we then show that $\|X_t - x^\star\| \to 0$, which is a contradiction (we take $r$ small enough so that $x^\star$ is the only solution of (Opt) in $\mathbb{B}_r(x^\star)$). We have therefore proved that $\|X_t - x^\star\| \to 0$ for any realization of $\mathcal{E}$. In conclusion, $X_t$ converges to $x^\star$ with probability at least $1 - \delta$. $\qquad\square$

## F.5 Proof of Proposition 2

**Proposition 2.** *If a solution $x^\star$ satisfies Assumption 5', then for every $\varepsilon > 0$, there is a neighborhood $U$ of $x^\star$ such that the error bound condition (EB) is satisfied on $U$ with constant $\tau = \sigma_{\min} - \varepsilon$ where $\sigma_{\min}$ denotes the smallest singular value of $\mathrm{Jac}_V(x^\star)$.*

*Proof.* By definition of Jacobian we have

$$
V(x) = V(x^\star) + \mathrm{Jac}_V(x^\star)(x - x^\star) + o(\|x - x^\star\|). \tag{F.15}
$$

By the min-max principle of singular value it holds

$$
\|\mathrm{Jac}_V(x^\star)(x - x^\star)\| \geq \sigma_{\min} \|x - x^\star\|. \tag{F.16}
$$

Since $V(x^\star) = 0$, combining (F.15) and (F.16) gives

$$
\|V(x)\| \geq \sigma_{\min} \|x - x^\star\| - o(\|x - x^\star\|).
$$

We conclude by noticing $\mathrm{dist}(x, \mathcal{X}^\star) = \|x - x^\star\|$ when $U$ is small enough. $\qquad\square$

## F.6 Proof of Theorem 4

**Theorem 4.** *Fix a tolerance level $\delta > 0$ and suppose that Assumptions 1'–3' and 5' hold for some isolated solution $x^\star$ of (Opt) with $q > 3$. Assume further $x^\star$ satisfies Assumption 5' and (DSEG) is run with stepsize parameters of the form $\gamma_t = \gamma/(t+b)^{1/3}$ and $\eta_t = \eta/(t+b)^{2/3}$ with large enough $b, \eta > 0$. Then, there exist neighborhoods $U, U'$ of $x^\star$ and an event $E_U$ such that:*

*a)* $\mathbb{P}(E_U \mid X_1 \in U) \geq 1 - \delta$.
*b)* $\mathbb{P}(X_t \in U' \text{ for all } t \mid E_U) = 1$.
*c)* $\mathbb{E}[\|X_t - x^\star\|^2 \mid E_U] = \mathcal{O}(1/t^{1/3})$

*In words, if (DSEG) is not initialized too far from $x^\star$, the iterates $X_t$ remain close to $x^\star$ with probability at least $1 - \delta$ and, conditioned on this event, $X_t$ converges to $x^\star$ at a rate $\mathcal{O}(1/t^{1/3})$ in mean square error.*

*Proof.* Both a) and b) are direct consequences of Theorem F.1. In effect, since $q > 3$, the sum of the series $\sum_t \eta_t^2$, $\sum_t \gamma_t^2 \eta_t$ and $\sum_t \gamma_t^q$ can be made arbitrarily small by taking sufficiently large $b$. Moreover, $x^\star$ is an isolated solution because $\mathrm{Jac}_V(x^\star)$ is non-singular. Therefore, taking $E_U := E_\infty^\rho$, $U := U^\rho$ and $U' := \mathbb{B}_{\rho r}(x^\star)$ readily gives a) and b).

Finally, to guarantee c), we need to have $\rho$ small enough and enforce $\gamma \eta \sigma_{\min}^2 (1 - \gamma_1 \beta) > 1/6$. In fact, from $\gamma \eta \sigma_{\min}^2 (1 - \gamma_1 \beta) > 1/6$ we deduce the existence of $\varepsilon \in (0, \sigma_{min})$ such that $\gamma \eta (\sigma_{\min} - \varepsilon)^2 (1 - \gamma_1 \beta) > 1/6$. Since $\mathrm{Jac}_V(x^\star)$ is non-singular, by Proposition 2 we can choose $\rho > 0$ so that

the error bound condition (EB) is satisfied on $\mathbb{B}_{\rho r}(x^\star)$ with $\tau = \sigma_{\min} - \varepsilon$. Let $\mathcal{M}_1, \mathcal{M}_2$ be defined as in Appendix F.4. We then obtained from (F.14)

$$\mathbb{E}[\|X_{t+1} - x^\star\|^2 \, \mathbb{1}_{E_{t+\frac{1}{2}}}] \leq (1 - 2\gamma_t \eta_t \tau^2 (1 - \gamma_t \beta)) \, \mathbb{E}[\|X_t - x^\star\|^2 \, \mathbb{1}_{E_{t+\frac{1}{2}}}] + \gamma_t^2 \eta_t \mathcal{M}_1 + \eta_t^2 \mathcal{M}_2.$$

By using $E_{t+\frac{1}{2}} \subset E_{t-\frac{1}{2}}$, we get

$$\mathbb{E}[\|X_{t+1} - x^\star\|^2 \, \mathbb{1}_{E_{t+\frac{1}{2}}}] \leq (1 - 2\gamma_t \eta_t \tau^2 (1 - \gamma_t \beta)) \, \mathbb{E}[\|X_t - x^\star\|^2 \, \mathbb{1}_{E_{t-\frac{1}{2}}}] + \gamma_t^2 \eta_t \mathcal{M}_1 + \eta_t^2 \mathcal{M}_2$$

Therefore, with the specified stepsize policy and the condition $\gamma \eta \tau^2 (1 - \gamma_1 \beta) > 1/6$, applying Lemma D.2 yields $\mathbb{E}[\|X_{t+1} - x^\star\|^2 \, \mathbb{1}_{E_{t+\frac{1}{2}}}] = \mathcal{O}(1/t^{1/3})$. Finally

$$\mathbb{E}[\|X_t - x^\star\|^2 \mid E_\infty^\rho] = \frac{\mathbb{E}[\|X_t - x^\star\|^2 \, \mathbb{1}_{E_\infty^\rho}]}{\mathbb{P}(E_\infty^\rho)} \leq \frac{\mathbb{E}[\|X_t - x^\star\|^2 \mathbb{1}_{E_{t-\frac{1}{2}}}]}{1 - \delta},$$

which proves $\mathbb{E}[\|X_t - x^\star\|^2 \mid E_\infty^\rho] = \mathcal{O}(1/t^{1/3})$. $\qquad\square$