[Reviews · NeurIPS 2020]

Review 1

Summary and Contributions: Edit after author feedback: after reading other reviews and the feedback of the authors, I do not wish to modify my appreciation. ### The primary goal of this paper is to propose a variation of the extragradient method for saddle point problems with underlying vector field V. This method should accomodates noisy estimates in vector field measurement. While a direct adaptation of the extragradient method is proved not to converge to the solution set of the problem problem, the authors investigate the possibility to use two different regimes for both step sizes appearing in the extragradient algorithm. As suggested by the title, the update step size should satisfy a much stronger decrease condition compared to the second one. The authors first describe various intuitive ways to look at this idea and then propose a variety of theoretical results: global convergence under usual noise conditions, regularity and stability conditions for the variational inequality problem (such as extensions of monotonicity, error bound condition or affinity of the vector field). Furthermore the authors are able to prove local convergence results for isolated solutions of the variational inequality problem only requiring local version of the stability and regularity condition for the vector field. These results would in principle apply to "non convex non concave" min max problems as they only require local assumptions around isolated solutions. The results are illustrated on synthetic saddle point and machine learning problems.

Strengths: The paper is well written and makes a clear and convincing statement regarding the use of two step size regimes for the extragradient method in stochastic setting. Obtaining converging algorithms in such settings is already an important challenge and the author propose an important variety of results ranging from qualitative convergence analysis to convergence rate analysis under more stringent assumptions. Furthermore, the authors were able to devise local convergence results. While the formulation sounds a bit imprecise, these are very relevant to machine learning applications such as GAN's for which global stability properties do not hold. Global stability of stochastic algorithms constitutes a very challenging question. The theoretical developments are quite impressive to me and I do not doubt that these will be inspiring in many other machine learning applications where stability of stochastic approximation algorithms is an issue.

Weaknesses: While the authors focus on saddle point problems, it seems that the proposed results would apply to any variational inequality problem satisfying the desired stability and regularity conditions. While I understand the motivation, I think that this could be mentioned more explicitely. Regarding local convergence results, the main text assume that some of the assumptions hold locally. However it is not really clear what assumptions 2 and 3 mean locally. While the appendix provides a precise statement, I think it would be adequate to explicitely refer to it in the main text. Furhtermore, the statements are very vague ("proper choice" of constants, "small enough" constants, "close enough" initialization), this could be written in more rigorous mathematical terms.

Correctness: I did not have time to check in full details all results of appendix F, my knowledge of the involved techniques is actually quite limited. Despite this word of caution, the proof arguments are very convincing and up to my understanding correct.

Clarity: The paper is overall very clear, in terms of global presentation and articulation of the ideas, level of english language and precision of mathematical statements and proofs. I have a very positive view regarding this work.

Relation to Prior Work: There is a long discussion about connection to existing art in the introduction, which is completed in the appendix. I would not be able to ensure that this constitutes and exhaustive discussion but this is up to my knowledge a very detailed and complete presentation of related litterature.

Reproducibility: Yes

Additional Feedback: - Line 112, maybe put the nonemptyness of the solution set as a numbered assumption as it is never explicitely called in all results. - Line 120, what is the quantification on q? - Line 127 maybe define what is a Cartheodory function, or put a reference. - The discussion aroud assumption 4 and the begining of Section 5.1 are repetitions of the end of Section 4? this could be harmonized. - The "covariance matrix learning" problem is kind of a strange denomination. Covariance learning could be identified with the unrelated field of covariance matrix estimations. What the author propose is actually a simple GAN model and I think that it would be much clearer if the chosen denomination reflected it. Appendix: Line 440, maybe rephrase? I did not really understand the interest of Section B? Line 573, more precise quantification on x* would help understand the interplay between (1) and (2). Properly chosen and large enough on lines 608, 609 are vague and could be more explicit Line 670, I guess the authors sum starting from t=3. Line 702, maybe precise that the very definition of D_{t+1/2} allows to check half of the inequalities of Lemma F.1 and that with F.9 the required recursive inquality is checked with all half integers. Line 706 I did not really understand the relation between Gamma and gamma, the statement is about existence of Gamma, but line 706 says that it can be chosen small enough. Last pages seemed correct although I could only briefly check them.


Review 2

Summary and Contributions: The paper studies the stochastic extragradient method and in particular a special case of the classic algorithm where the exploration step evolves at a more aggressive time scale compare to the update step. The proposed method is called double stepsize extragradient algorithm (DSEG). The authors provide last-iterate convergence guarantees for DSEG under an error bound condition. --- Post rebuttal ---- I have read the authors' response, and I decided to change my score to "A good submission; accept". The paper is well written and clear and has the potentials to be a bedrock for some future considerations in the area. However I highly encourage the authors to update the section of the main contributions. In particular, (i) to clear the part with the counter-example and (ii) to compare DSEG (in theory and numerical experiments) with the algorithms proposed in [1] that solve essentially the same problems and come also with last-iterate convergence guarantees.

Strengths: The paper is really well written and the contributions are clear. To the best of my knowledge the double step-size idea for extragradient is new. The proposed theoretical analysis provide last-iterate convergence for the DSEG. This is of great importance especially considering that most guarantees on the classic extragradient method and its variants rely on iterate averaging over compact domains.

Weaknesses: The idea of the paper is simple and the motivation of the approach is well justified. However i have some concerns on the novelty of the work and the comparison with existing results. In particular I have the following concerns: Main Issues: I am concern regarding two of the main contributions of this work. I believe that some of the claims on the main contributions section are not novel. 1) The fact that the stochastic EG did not converge was shown with a counterexample in Chavdarova et al. [2]. However, in line 82, the authors mentioned that their approach on counterexample is an improvement because they show that the non-convergence persists for any error distribution with positive variance. In my opinion the proposal counterexample is not really a main contribution of this work. This work builds upon the counterexample of Chavdarova et al. [2] and use it to propose a convergent variant of the EG. 2) In line 55 the authors claim: "Prior to our work, last-iterate convergence rate for bilinear min-max games had only been studied in the deterministic setting." This is not true. In ICML 2020 Loizou et. al proposed the analysis of stochastic Hamiltonian methods showing last-iterate convergence for stochastic bilinear games and some classes of non-convex non-concave games. They also proposed a variance reduced method showing linear convergence (which is much faster then the O(1/t) rate of the DSEG in this setting). Other Issues: 3) The authors mentioned that the error bound condition is satisfied for two large classes of problems: Strongly monotone operators and Affine operators. However the definitions of the above two problems were never explicitly given. In addition for the affine operators it is mentioned that $\tau$ is the minimum non-zero singular value of the matrix. Which matrix? the authors need to be more rigorous. 4) In line 116 the authors start using capital letters for the notation (see $X_t$). until this point everything was lower case. 5) In experiments the only method presented is the DSEG with different values for $r_\gamma$ and $r_eta$. This is not adequate. The method should compare with other methods that guarantee convergence for the classes of games under study, like the stochastic Hamiltonian methods mentioned above. Even if the DSEG will be slower it will be really helpful for the reader. In addition the indicators of the lines in the Figure 3 are not distinguished (for colour-blind readers)

Correctness: I check the important parts of the proofs and the derivation and all the proofs seem correct.

Clarity: The paper is well written and clear.

Relation to Prior Work: Closely related paper: [1] Loizou, Nicolas, Hugo Berard, Alexia Jolicoeur-Martineau, Pascal Vincent, Simon Lacoste-Julien, and Ioannis Mitliagkas. "Stochastic Hamiltonian Gradient Methods for Smooth Games." ICML 2020

Reproducibility: Yes

Additional Feedback:


Review 3

Summary and Contributions: This paper studies large-scale saddle-point problems in machine learning. The authors proposed a stochastic extragradient method that allows different and variable stepsize scaling. The proposed algorithm, named DSEG, suggests a more aggressive time-scale of the exploration step compared to the update step. Convergence analysis and numerical studies were implemented to justify the proposed method.

Strengths: The variable stepsize scaling idea is interesting and its advantage over equal scaling was explained clearly in the paper. The presentation of the paper is smooth. The convergence of the proposed method is carefully analyzed.

Weaknesses: According to (4), DSEG introduces 5 parameters (or 3 in the case of Thm 4) to tune the variable stepsize scaling. In either case, the paper solves the tuning parameter selection problem by involving more parameters. The paper did not provide a careful discussion of the tuning parameter selection issue. This may impede the potential users to apply DSEG. Besides, maybe due to the limitation of the space, the numerical experiments are not comprehensive to support the arguments developed in the paper.

Correctness: The proposed method is not complicated and the intuition seems reasonable. The condition ``locally near some isolated solution" in Theorem 2 was not very clearly defined. Assumption 2(a) requires the gradient estimator to be (conditionally) unbiased which may not be practical in many applications. The numerical experiments in the main document are brief with many replication details missing.

Clarity: The presentation of the paper is smooth though there are some grammar errors.

Relation to Prior Work: The paper has made good connection between the proposed method and prior work in the field. The motivation and advantage of variable stepsize scaling is clearly explained.

Reproducibility: No

Additional Feedback: ######## EDIT after author's response ############# I have carefully read the rebuttal letter. Though I am not fully satisfied with the authors' responses, I would like to raise my score to 6. ######## Original response ############# Please see my comments above.


Review 4

Summary and Contributions: The paper seeks to understand why extra-gradient (EG) algorithms, which often work well in finding local minima, commonly fail when dealing with saddle-point problems. Where EG algorithms do work, the equivalent stochastic EG algorithm may fail. This is import for a number of reasons, not least due to the widespread use of GANs. To this end, the paper first proves a result demonstrating that stochastic EG does not converge even on bilinear problems. The paper then looks instead at a double step-size extra-gradient algorithm (DSEG) and proves that it can converge on said bilinear problems, provided the two step-sizes are not equal (cf. the EG algorithm). This serves as the motivation for a more general analysis of the global and local convergence of DSEG. The paper concludes that a more aggressive exploration step and a less aggressive exploration step in DSEG helps prevent the problems found in EG algorithms. In particular, the paper makes the following substantial contributions: 1) Proves convergence of DSEG under reasonable and relatively mild assumptions (corresponding to a substantial range of problems). 2) Finds DSEG convergence rates for both constant and decreasing step-sizes. In particular, it determines how quickly step-sizes can be decreased to still allow convergence and what the decrease rate leading to fastest convergence is. 3) Obtains sharper convergence rates under the assumption that the vector field associated to the optimization problem is affine. Finally, the paper includes experimental validation of the theoretical results and a discussion of how the optimistic gradient method may benefit from similar step-size modifications.

Strengths: The paper is easy to read, well-motivated, explains its relation to earlier work, provides a thorough theoretical analysis, and contains appropriate experiments. All in all, a clear accept.

Weaknesses: This was a strong paper with no major problems.

Correctness: The motivation and theoretical analysis were logical and sound.

Clarity: The paper was well-written and easy to follow throughout. There were no obvious typos, notational errors, or formatting problems.

Relation to Prior Work: Papers published on similar topics were mentioned, and the differences between those and this paper were made clear.

Reproducibility: Yes

Additional Feedback: ### Post-author feedback edit ### I've read the response from the authors and the points raised by the other reviewers. I am keeping my original rating.

[Author Response · NeurIPS 2020]

We thank all the reviewers for their valuable comments and overall positive feedback. We will fix typos and improve our presentation in the next version.

**Reviewer #1:**

Thank you very much for your careful reading and recognition our work. We will make modifications to the paper as suggested. In particular, we will provide the precise statements of the local assumptions in the main text if space permits and replace the qualitative description in our proof (e.g., large enough, proper choice) by formal mathematical statement whenever possible. We will also explain that the so-called covariance learning problem here is just a very simple GAN model to avoid confusion. The other points will be addressed similarly. Concerning the quantity $\Gamma$: we mean that when $\Gamma$ is small enough those inequalities are verified and therefore there exists such $\Gamma$ satisfying the desired properties. Finally, the goal of Appendix B is to show that the same conclusion may hold for optimistic gradient as well by adequately choosing the output vector. We also attempt to give an explanation of some experimental results observed in the paper of Ryu et al. (2019) through the lens of DSEG.

**Reviewer #2:**

1. In terms of the relation of our counterexample to the one of Chavdarova et al. (2019): The purpose of this counterexample was to motivate the analysis to come, it was not intended as a separate contribution, nor labeled as such. We will discuss in more detail the exact differences between our setting and that of Chavdarova et al.

2. Regarding the paper of Loizou et al. (2020): We thank the reviewer for pointing us to this paper. At the same time, we would like to point out that this paper was accepted to ICML in June 2020, and it only appeared on arxiv on July 8, 2020. We will be happy to cite and discuss it, but mentioning it as a "weakness" is unfair: our work cannot be compared to a paper that was not even available on arxiv at the time of abstract registration at NeurIPS.

3. Strong monotonicity is a standard notion, but we will provide a reference to the standard textbook of Facchinei-Pang (as we would like to avoid introducing definition that are not explicitly used in the results). As for the definition of affine operator and its associated matrix, we will add a footnote about it since most readers are likely familiar with this notion.

4. About the notations: A capital $X$ is used to mark a state of an algorithm (a stochastic process, so typically noted with uppercase letters) while a lower case $x$ represents a point in the space.

5. Thank you for your suggestion of comparing against other methods. We will run the corresponding experiments and discuss the differences between different algorithms. We will also update the plot for colorblind readers.

**Reviewer #3:**

Hyperparameter tuning is a recurring problem in machine learning. Our goal was not to provide a parameter-free method, but a method with superior convergence guarantees. Developing an adaptive, parameter-free variant is an entire new paper by itself. Our paper provides the bedrock for this (and other) future considerations, a fact which highlights the potential impact of our work. As for the numerical experiments, since our paper is mainly theoretical, experimental details were relegated to the appendix, where we provide all the necessary reproducibility material. Concerning what we mean by a localized version of the assumptions: it could indeed be difficult to grasp for many readers. While in the submitted version the formal statements of these local assumptions appear in the appendix, we plan to move them to the main text in a newer version. Finally, the unbiasedness of the operator is a standard assumption in the theoretical analysis of stochastic optimization methods and is also verified in most machine learning applications (e.g., mini-batch setting where the estimator is unbiased by construction). Bias only appears to be an issue in very specific applications.

**Reviewer #4:**

Thank you very much for your careful reading, comments and appreciation of our work.

# References

Chavdarova, T., Gidel, G., Fleuret, F., and Lacoste-Julien, S. Reducing noise in gan training with variance reduced extragradient. In *NeurIPS*, 2019.

Loizou, N., Berard, H., Jolicoeur-Martineau, A., Vincent, P., Lacoste-Julien, S., and Mitliagkas, I. Stochastic hamiltonian gradient methods for smooth games. In *ICML*, 2020.

Ryu, E. K., Yuan, K., and Yin, W. ODE analysis of stochastic gradient methods with optimism and anchoring for minimax problems and GANs. `https://arxiv.org/abs/1905.10899`, 2019.


[Meta-Review · NeurIPS 2020]

Dear authors, Thank you for submitting your clear and well-written paper. I am pleased to report that all reviewers liked your paper and see a potential for the field. When producing the final camera-ready paper, please check the reviewer's remarks to make it stronger. Thank you